# ANCHOLIK-NER: A benchmark dataset for Bangla regional named entity recognition

**Bidyarthi Paul**[1], **Faika Fairuj Preotee**[1], **Shuvashis Sarker**[1], **Shamim Rahim Refat**[2], **Shifat Islam**[3], **Tashreef Muhammad**[1]*, **Mohammad Ashraful Hoque**[1], **Shahriar Manzoor**[1]

**1** Department of CSE, Southeast University, Dhaka, Bangladesh, **2** Department of CSE, Ahsanullah University of Science and Technology, Dhaka, Bangladesh, **3** Department of CSE, Bangladesh University of Engineering and Technology, Dhaka, Bangladesh

* tashreef.muhammad@seu.edu.bd

## Abstract

Named Entity Recognition (NER) in regional dialects is a critical yet underexplored area in Natural Language Processing (NLP), especially for low-resource languages like Bangla. While NER systems for Standard Bangla have made progress, no existing resources or models specifically address the challenge of regional dialects such as Barishal, Chittagong, Mymensingh, Noakhali, and Sylhet, which exhibit unique linguistic features that existing models fail to handle effectively. To fill this gap, we introduce ANCHOLIK-NER, the first benchmark dataset for NER in Bangla regional dialects, comprising 17,405 sentences and 101,817 words annotated with 10 entity tags across 5 regions. The dataset was sourced from publicly available resources and supplemented with manual translations, ensuring alignment of named entities across dialects. We evaluate three transformer-based models—Bangla BERT, Bangla Bert Base, and BERT Base Multilingual Cased—on this dataset. Bangla BERT achieved the highest performance overall, with F1-scores of 82.27% (Mymensingh), 81.48% (Barishal), 78.75% (Sylhet), 78.50% (Noakhali), and 75.31% (Chittagong). These results highlight strong recognition capability in Mymensingh and Barishal, while dialectal variation in Chittagong remains challenging. As no prior NER resources exist for Bangla regional dialects, this work provides a foundational dataset and baseline benchmarks to facilitate future research. Future work will focus on dialect-aware model adaptation and expanding coverage to additional regions.

## Introduction

Named Entity Recognition (NER) is a part of Information Extraction (IE) that focuses on identifying and classifying named entities in unstructured text – names of persons, organizations, locations, dates, numbers and other specific terms. By extracting entities, Named Entity Recognition (NER) transforms raw text into structured data, enabling tasks like information retrieval, question answering, and knowledge graph

**Data availability statement:** All Dataset files are available from the Zenodo database (accession number(s) 17806539), https://zenodo.org/records/17806539.

**Funding:** This study was funded by Institute of Research and Training, Southeast University, Bangladesh. Hence, we had financial support. Southeast University, Bangladesh (SEU/IRT/RG/2025/01/09 to M.A.H.).

**Competing interests:** The authors have declared that no competing interests exist.

construction. NER was first introduced in the Sixth Message Understanding Conference (MUC-6) which emphasized extracting structured information from free-form natural language [1]. Standard entity types like Person, Organization and Location were defined in MUC-6 and later refined in MUC-7 with clearer annotation guidelines and evaluation metrics [2]. Multilingual adaptation efforts like The Conventions d'annotations en Entités Nommées-ESTER project addressed language specific annotation conventions [3]. Today NER is a foundation step in NLP pipelines supporting applications like machine translation, automatic summarization, topic detection and recommendation systems [4]. So NER is essential for systems that want to extract semantic meaning and support decision making in both academic and industrial NLP tasks.

Initial approaches to the NER task were rule-based, relying on manually crafted lexicons, grammars and pattern-matching heuristics to extract entities from domain-specific corpora [5–7]. These systems were limited in scalability, language coverage and generalizability as they were dependent on domain knowledge and rigid linguistic rules. As the limitations of handcrafted systems became apparent—especially in adapting to different languages and sentence structures—researchers moved towards data-driven approaches. This led to the adoption of statistical and machine learning models that required annotated datasets but offered much better adaptability and inference capabilities. Among the earliest machine learning models used for NER were Hidden Markov Models (HMMs), which modelled sequences of labels with probabilistic transitions and emissions [8,9]. These were soon followed by more powerful models like Conditional Random Fields (CRFs) [10,11] which could capture both contextual features and label dependencies across sequences. In parallel, Maximum Entropy (ME) [12] and Maximum Entropy Markov Models (MEMMs) [13] were explored for their flexibility in incorporating different features. Support Vector Machines (SVMs) [14,15] were introduced as robust classifiers for NER, especially when combined with rich feature representations and kernel methods. To further improve performance, Ensemble Modeling techniques [16,17] were proposed, combining multiple learners to mitigate overfitting and improve generalization. With the increasing availability of annotated NER datasets across multiple languages, multilingual and cross-lingual models started to emerge [18] allowing knowledge transfer between related language families. The evolution of NER reached a new milestone with the rise of deep learning, especially recurrent neural networks (RNNs) and their variants like Long Short-Term Memory (LSTM) networks. These models could capture long-range dependencies in text and when combined with CRFs produced state-of-the-art results in sequence labelling tasks [19]. The introduction of Bi-directional LSTM-CRF architectures further improved entity recognition by leveraging both past and future context [20]. The most significant breakthrough came with the development of pre-trained transformer-based models like BERT and multilingual BERT (mBERT) [21] which enabled deep contextualized word embeddings and fine-tuning for downstream tasks including NER. Variants like RoBERTa [22], and XLM-R [23] for low-resource and performance across languages. In recent years, NER esearch has continued to evolve rapidly. GLiNER [24], a lightweight generative NER framework, outperformed ChatGPT in zero-shot evals and used way less memory.

ToNER [25], introduced type-oriented generation and showed great results in prompt-based NER. KoGNER [26] added knowledge graph distillation to transformer-based models and improved entity classification in biomedical NER. Few-TopNER [27] proposed a few-shot, topic-aware method and improved cross-lingual generalization for low-resource NER. For noisy and informal data like social media, SEMFF-NER [28] used multi-scale feature fusion and syntactic information to handle irregular sentence structures better. Large Language Model Cooperative Communication (LLMCC) [29] tried cooperative learning between large language models for NER and got better extraction consistency through model coordination. Named Entity Recognition for All (NER4All) [30] addressed challenges in historical and multilingual corpora and introduced methods for adaptive entity recognition across genre and domain shifts. Multimodal NER [31] also made progress with Adaptive Mixup methods that fused image-text representations for better recognition in visually grounded contexts. These recent developments demonstrate that NER continues to advance through interdisciplinary approaches and scalable modeling frameworks.

In recent years, a lot of progress has been made in Bangla Named Entity Recognition (NER) systems which is one of the most spoken yet under-resourced language in the world. Initial works used statistical models like Conditional Random Fields (CRFs) and Support Vector Machines (SVMs) with handcrafted features on small scale annotated corpora [32]. Later Bi-LSTM and Bi-LSTM-CRF were introduced which offered better contextual modeling and yielded better results [33]. With the rise of pre-trained language models, transformer based approaches like multilingual BERT (mBERT) and BanglaBERT were used for Bangla NER and further improved the results by fine-tuning on domain specific tasks [34]. Along with modeling techniques, several tools and resources were developed including BNLP toolkit [35], BnNER and BNERCorpus [36]. Recently B-NER released one of the largest Bangla NER dataset to date along with benchmark evaluations [36]. Gazetteer-Enhanced improved the recognition performance by using K-Means infused CRF with BanglaBERT embeddings [37]. TriNER [38] explored multilingual NER by training models across Hindi, Bengali and Marathi and enabled cross-lingual learning for low resource languages. Despite all the progress in Named Entity Recognition (NER) for Bangla, there are still many limitations that hinder the development of high performing and generalizable models. Most existing Bangla NER models are trained on small or synthetic datasets that fail to capture the linguistic diversity and naturally occurring variation of the language. As a result, these models perform poorly on diverse contexts like informal speech, domain specific corpora and most critically regional dialects. One of the most overlooked yet impactful challenge in Bangla NER is the lack of support for regional dialects which are widely spoken in Bangladesh in areas like Chittagong, Sylhet, Barishal, Khulna and Mymensingh. These dialects vary greatly from standard Bangla in terms of vocabulary, grammar, pronunciation and semantics making it difficult for models trained only on formal Bangla corpora to identify and classify named entities in dialectal text. This gap is particularly problematic for practical applications like social media analysis, regional news summarization, public health communication and localized digital services – domains where dialectal usage is frequent and essential. Moreover, the absence of large scale, gold standard annotated corpora along with inconsistencies and mismatches in translation based datasets further restricts model performance. Transformer based models like BanglaBERT and mBERT, although promising, still lag behind those for high resource languages due to limited morphological, cultural and linguistic adaptation. The exclusion of dialectal diversity not only leads to poor recognition and inaccurate information extraction but also introduces a form of linguistic bias that undermines inclusivity. Given the socio-cultural importance of regional speech in Bangladesh, development of dialect aware NER models is not merely a technical improvement but also a linguistic and ethical imperative. Addressing these challenges requires a systematic effort to build large, high quality and regionally diverse datasets along with adaptable and context sensitive modeling approaches that can serve all segments of Bangla speaking populations.

Creating a comprehensive and dialect-aware Bangla NER dataset is therefore a crucial step toward enabling broader linguistic and technological advancements beyond entity recognition itself. Such a dataset not only supports the development of robust NER models but also lays the foundation for several downstream Natural Language Processing (NLP) tasks that depend heavily on accurate language understanding. For instance, regional dialect data can significantly improve machine translation systems by facilitating more accurate Standard Bangla to regional dialect translations and

vice versa [60,71]. Similarly, access to annotated dialectal corpora can enhance dialect identification models [72], allowing systems to automatically detect and adapt to regional variations in text. Furthermore, in tasks such as emotion classification [73] and hate speech detection [74], dialect-sensitive representations can reduce misclassification caused by regional word usage, idioms, or informal expressions common in social media and community discourse. By capturing the richness and diversity of regional Bangla, this dataset can thus serve as a linguistic bridge between standard and non-standard varieties—enabling fairer, more inclusive, and contextually aware NLP systems in the future.

To realize these opportunities, our approach takes a data centric view and emphasizes linguistic inclusivity and technical precision. Rather than focusing on model architecture or fine tuning strategies we focus on enhancing the quality, diversity and dialectal relevance of the training data used in Bangla NER. While previous works have focused on standard Bangla we explore underrepresented regional linguistic contexts by curating dialect aware datasets and translation corpora that preserve named entity integrity. We investigate how dialectal variation affects NER and how it can be modeled through better data design. Fig 1 illustrates an example of Dialect variations in Bangla language in NER. We also address the challenges in translation based NER where entity misalignments and inconsistencies degrade the model performance by incorporating structured correction and validation strategies. Our approach is based on empirical benchmarking using pre-trained transformer models so that each proposed resource is practical and measurable in terms of downstream effectiveness. Here are the key contributions of our work:

- **Regional Bangla Dialect-Aware NER Dataset**
  We present the first NER dataset for Bangla that includes annotations from five major regional dialects—*Chittagong, Sylhet, Barishal, Noakhali*, and *Mymensingh*—addressing the lack of dialectal coverage in existing resources.
- **Hybrid Translation Dataset with Entity Alignment**
  We construct a parallel corpus between standard Bangla and regional dialects, ensuring that *named entities* are consistently preserved and aligned across translations.
- **Anomaly Detection and Data Refinement**
  We apply systematic anomaly detection and correction to remove noisy or inconsistent data, ensuring high-quality hybrid datasets for robust model training.

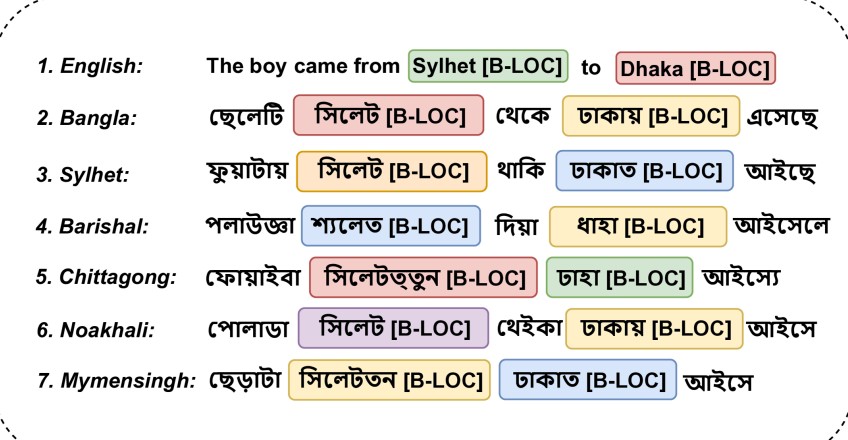

**Fig 1**. **Regional NER examples along with Standard Bangla and English.**

- **Benchmarking with Established Transformer Models**
  To provide reproducible baselines, we evaluate the proposed dataset using established transformer architectures—**Bangla BERT**, **Bangla Bert Base**, and **BERT Base Multilingual Cased**. This allows us to quantify model robustness across dialectal variations without introducing architectural modifications.

By addressing critical challenges in Bangla NER—including dialectal diversity, limited annotated resources, and translation inconsistencies—our approach ensures better generalization, higher entity recognition accuracy, and improved adaptability across regional variations. This results in an adaptable NER framework applicable to diverse Bangla language contexts. The paper is structured as follows. The **Problem description** section describes the problem in detail, highlighting the core challenges addressed in this study. The **Background** section presents a comprehensive review of existing research and methodologies in Bangla NER. The **Dataset preparation** section details the dataset construction process, including data collection as well as data pre-processing and tokenization strategies used to curate and refine the datasets. The **Methodology** section outlines the proposed framework, including dataset construction, anomaly detection techniques, and benchmarking approaches. The **Result analysis** section presents the performance evaluation and experimental analysis of the proposed methods. Finally, the **Conclusion and future work** section summarizes the key findings, discusses the limitations of the study, and outlines directions for future research.

## Problem description

Let $n$ represent the number of words in a sentence. The sentence can be denoted as a set $s = \{w_1, w_2, w_3, \dots, w_n\}$, where each word is treated as an individual token. The output is a corresponding set of tags $t = \{t_1, t_2, t_3, \dots, t_n\}$, where each tag $t_i$ belongs to the set of predefined entity tags:

$$t_i \in \{\text{B-PER, I-PER, B-LOC, I-LOC, B-ORG, I-ORG, B-REL, I-REL, B-FOOD,}$$
$$\text{I-FOOD, B-ANI, I-ANI}, O, \text{B-COL, I-COL, B-ROLE, I-ROLE, B-OBJ, I-OBJ}\}.$$

The context of the sentence must be considered when assigning tags to each token.

## Related works

With the rapid expansion of user-generated content, social networking platforms, and digital services, Named Entity Recognition (NER) has become a critical task in the field of Natural Language Processing (NLP), serving as a foundational step in information extraction pipelines. NER facilitates structured understanding of text by identifying key entities such as people, places, organizations, and temporal expressions—playing a vital role in applications like news aggregation, knowledge graph construction, recommendation systems, and biomedical research. In high-resource languages such as English, Spanish, and German, the combination of large annotated corpora, linguistically rich resources, and transformer-based models like BERT and RoBERTa has enabled researchers to achieve state-of-the-art F1 scores nearing 95%. However, this success has not translated evenly across languages. Low-resource languages, including Bangla, suffer from a scarcity of labeled datasets, inconsistent annotation schemes, and limited availability of pretrained models tailored to linguistic nuances. Moreover, existing Bangla NER systems often focus on coarse-grained entities (like person or location), neglecting fine-grained and domain-specific categories. Recent efforts have started addressing these gaps by developing domain-adaptable NER systems.

Recent advancements in Named Entity Recognition (NER) for low-resource languages and domain-specific applications reveal a shared focus on large-scale dataset creation, domain-adaptive pretraining, and leveraging large language models (LLMs). In the context of Chinese NER, Yao et al. [39] introduced AgCNER, a substantial annotated dataset for agricultural diseases and pests, alongside AgBERT, a domain-specific model achieving an impressive 94.34%

F1-score. Complementing this, researchers [40] constructed Chinese NER datasets from Internet novels and financial reports, adopting a semi-supervised, entity-enhanced BERT pretraining approach that integrated lexicon-level knowledge into deep contextual embeddings. Both studies demonstrate the critical role of large annotated datasets and domain-specific adaptations for enhancing NER across diverse Chinese text genres. A similar pattern emerges in Indonesian NER research, where Yulianti et al. [41] and Khairunnisa et al. [42] explored distinct but complementary approaches. The authors [41] developed IndoLER, a legal-domain NER dataset with 1,000 annotated court documents and 20 fine-grained entity types, demonstrating that transformer-based models like XLM-RoBERTa and IndoRoBERTa outperformed traditional architectures(BiLSTM-CRF), achieving F1-scores up to 0.929. In contrast, Khairunnisa et al. [42] introduced IDCrossNER, a cross-domain Indonesian dataset derived via semi-automated translation from English, and applied GPT-based augmentation and transfer learning, significantly improving NER performance in limited-data scenarios. These studies highlight the role of multilingual resources and domain adaptation in addressing Indonesian NER challenges. In the medical NER domain, spanning across Chinese, Bangla, and English-language social media, the focus shifts toward extracting structured health information from unstructured text. Ge et al. [43] introduced Reddit-Impacts, a novel dataset capturing clinical and social impacts of substance use from Reddit posts, annotated with 30 low-frequency but critical entity types. Their study demonstrated that few-shot learning models like DANN and GPT-3.5 outperformed traditional methods such as BERT and RoBERTa, highlighting the challenges of sparse entity recognition in health discourse. Aligning with this, Muntakim et al. [44] developed BanglaMedNER, the largest Bangla medical NER corpus with over 117,000 tokens across three key categories (Chemicals & Drugs, Diseases & Symptoms). Their BiLSTM-CRF model achieved a 75% macro F1-score, demonstrating the feasibility of applying deep learning techniques in low-resource, specialized medical contexts.

In multilingual settings, cross-lingual transfer has proven effective, as seen in NaijaNER [45], which covered five Nigerian languages and demonstrated that multilingual models outperform language-specific counterparts. This trend extends to Indo-Aryan languages, where annotated datasets for Assamese [46] and Purvanchal languages (Bhojpuri, Maithili, Magahi) [47] aligned entity labels with Hindi NER corpora, facilitating knowledge transfer among related languages. Such alignment not only enriches low-resource datasets but also enhances model adaptability across linguistically similar regions. Similar strategies have been applied to West Slavic languages like Upper Sorbian and Kashubian, leveraging Czech and Polish corpora for cross-lingual learning [48], and in Uzbek, where annotated datasets with BIOES tagging schemes improved entity boundary detection in legal texts [49]. In the medical domain, cross-domain adaptation and linguistic diversity remain critical. An annotated corpus of over 117,000 tokens for medical NER (MNER) [44] exemplifies structured dataset creation, while another study [50] addressed informal health discourse in Consumer Health Questions (CHQs), capturing dialectal variations and achieving a modest F1-score of 56.13% with BanglishBERT. Collectively, these works reflect a unified strategy of leveraging cross-lingual alignment, shared annotation schemes, and domain-aware embeddings to advance NER performance in low-resource and domain-specific contexts.

Although substantial progress has been made in NER in terms of dataset quality, annotation consistency, and domain adaptability. A challenge is observed in Urdu NER, where Anam et al. [51] highlighted the scarcity of annotated resources and addressed out-of-vocabulary (OOV) issues by leveraging FastText and Floret embeddings with RNN variants. Their approach successfully captured sub-word patterns, yet the dependence on benchmark datasets restricts model adaptability across broader domains. In specialized areas like biomedical NER, the problem of limited labeled data becomes even more pronounced. Gao et al. [52] tackled this by adopting a transfer learning and self-training framework, pretraining models on extensive biomedical corpora like SemMed and MedMentions. While their approach reduced the dependency on manual annotations, it also highlighted the reliance on domain-specific corpora for maintaining high performance. Extending further, Yan et al. [53] addressed the complexity of Chinese medical NER by integrating RoBERTa-wwm-ext, BiLSTM, multi-head attention, and a gated context-aware mechanism (GCA) to effectively model long-range dependencies and entity relationships. However, their evaluation on datasets like MCSCSet and CMeEE also pointed out challenges in achieving consistent performance across different medical subdomains.

A few research works have significantly contributed to the development of Bangla NER datasets, though challenges related to entity diversity, annotation quality, and model adaptability persist. For instance, Lima et al. [54] developed a large-scale Bangla NER dataset containing over 1 million tokens across six entity types; however, despite its size, the dataset struggled with class imbalance, particularly dominated by non-entity tokens. To mitigate this, the authors integrated character-level embeddings within a hybrid CNN-BiLSTM/GRU-CRF model, demonstrating improved recognition of complex entity patterns in morphologically rich Bangla text. Karim et al. [33] initiated this line of research by creating a dataset of over 71,000 sentences, annotated across four core entity types—person, location, organization, and object—using the IOB tagging scheme. Their model, which integrated Densely Connected Networks (DCN) with BiLSTM for sequence labeling, achieved an F1-score of 63.37%. Building on the need for improved datasets and architectures, Rifat et al. [55] expanded the scope by annotating 96,697 tokens and benchmarking several deep learning models, including BLSTM+CNN and BGRU+CNN. Their work marked a shift away from traditional machine learning models (e.g., HMM, CRF, SVM) towards neural architectures that leveraged character-level embeddings. Despite this, their best-performing model (BGRU+CNN) attained an F1-score of 72.66%, though annotation inconsistencies and dataset imbalance remained key limitations.

Addressing these shortcomings, Haque et al. [36] introduced B-NER, which not only increased annotation reliability but also broadened entity diversity to eight categories, including complex entities like events and artifacts. Unlike previous works, their dataset of 22,144 sentences was fully manually annotated, achieving a Kappa score of 0.82, ensuring high inter-annotator agreement. This rigorous annotation framework, combined with evaluations using IndicbnBERT, yielded a Macro-F1 score of 86%, significantly outperforming earlier models and datasets.

## Dataset preparation

Natural Language Processing (NLP) has emerged as a foundational area within artificial intelligence, aiming to equip machines with the ability to understand, interpret, and generate human language. Among the various tasks in NLP, Named Entity Recognition (NER) remains particularly challenging. As a sequence labeling problem, NER involves detecting and classifying named entities—such as persons, locations, organizations, and more—within unstructured text. The availability of large, high-quality annotated datasets has significantly advanced NER systems in high-resource languages.

In the context of Bangla, a language spoken by millions yet historically considered low-resource, there has been a noticeable shift in recent years. Efforts to improve Bangla NLP have led to the development of benchmark datasets for several key tasks, including abstractive text summarization [56], question answering [57], authorship classification [58], and machine translation [59]. These initiatives mark a promising step toward reducing the resource gap. However, NER-specific resources in Bangla remain limited, and existing datasets are primarily focused on Standard Bangla, neglecting the diverse range of regional dialects spoken across the country. Bangladesh is home to a range of regional dialects, including Sylhet, Chittagong, Barishal, Noakhali, and Mymensingh, each with its own distinct phonological, lexical, and syntactic characteristics. These dialects, while widely spoken, are significantly underrepresented in computational linguistics. The lack of annotated resources for these varieties presents a major obstacle for building inclusive and dialect-aware NLP systems. In the context of NER, this gap is particularly pronounced, as dialectal variations can alter named entity boundaries, spellings, and even categories, reducing the accuracy of models trained solely on Standard Bangla.

To address this gap, our work introduces ANCHOLIK-NER, the first benchmark dataset for Named Entity Recognition in Bangla regional dialects. This dataset is designed to support the development of NER models capable of understanding dialect-specific named entities across Sylhet, Chittagong, Barishal, Noakhali, and Mymensingh. The dataset construction process follows a systematic pipeline, ensuring high annotation quality through expert linguists familiar with these dialects. Fig 2, illustrates the pipeline for the development of our proposed dataset.

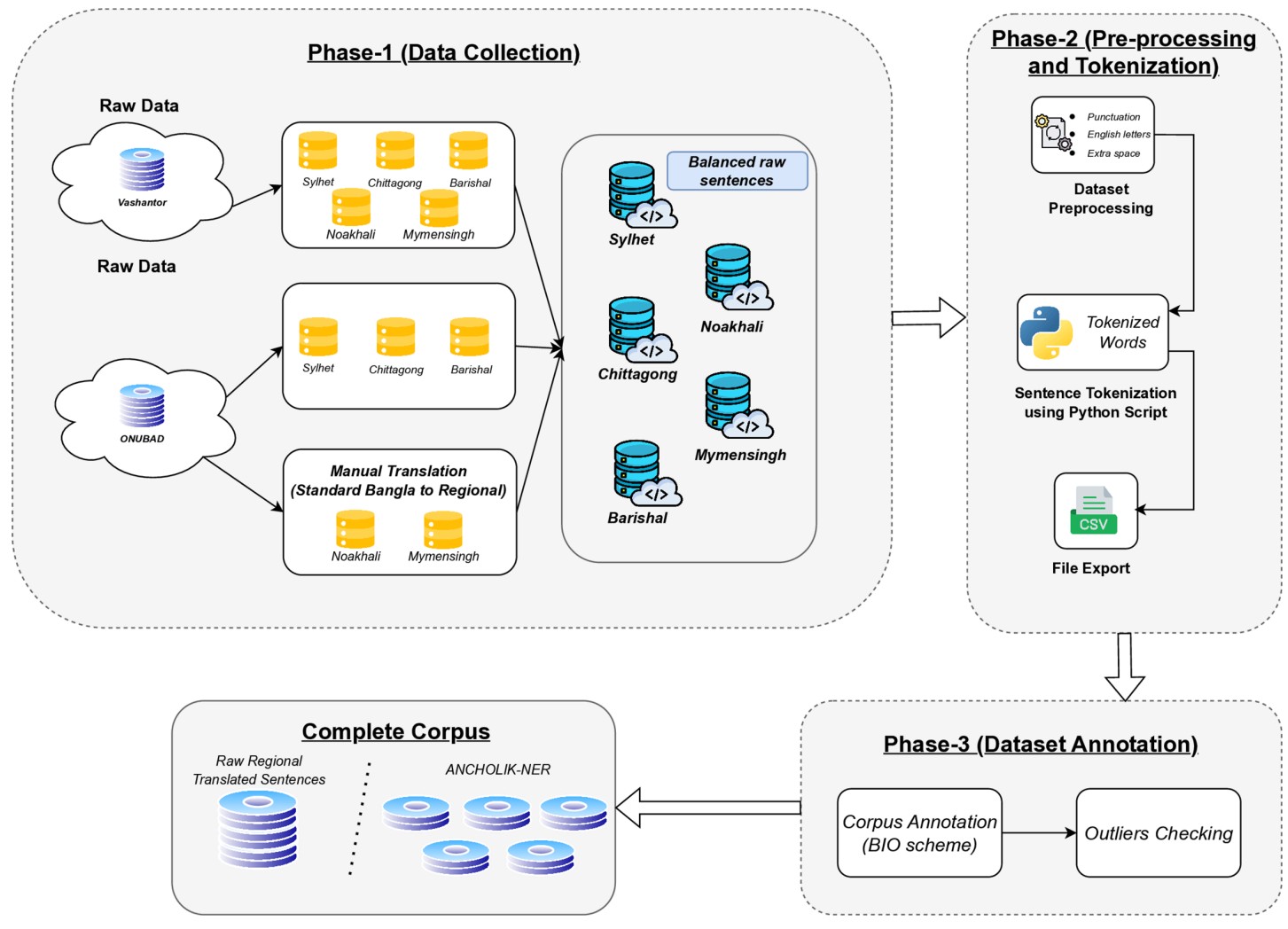

**Fig 2**. **Development of ANCHOLIK-NER: A systematic pipeline for dataset creation.**

## Data collection

Preparation of the ANCHOLIK-NER dataset starts with it's data collection phase, which was compiled from a combination of publicly available sources and manual collection efforts. A total of 17,405 sentences were gathered across five Bangla regional dialects: Sylhet, Chittagong, Barishal, Noakhali, and Mymensingh. The majority of the data—12,500 sentences—was sourced from the Vashantor corpus [60]. To construct a unified and consistent dataset, we merged the training, validation, and test splits from the original corpus into a single collection, while 2,940 sentences were extracted from the ONUBAD dataset [61].To enhance dialectal diversity and ensure balanced representation, an additional 1,965 sentences were manually obtained through regional dialect translation. As the ONUBAD dataset contains data for only three regions—Sylhet, Barishal, and Chittagong—we first collected the corresponding Standard Bangla versions of the sentences. These were then manually translated into the Noakhali and Mymensingh dialects to complete the dataset. Table 1, summarizes the distribution of the data sources.

**Table 1**. **Distribution of sentences across different data sources for Bangla regional dialects in the ANCHOLIK-NER dataset.**

| Sources | Sentence Count |
|---|---|
| Vashantor [60] | 12,500 |
| ONUBAD [61] | 2940 |
| Manual Translation | 1965 |
| **Total** | 17,405 |

## Data pre-processing and Tokenization

As the dataset was constructed by combining content from multiple sources—including existing datasets like Vashantor [60] and ONUBAD [61], as well as manually translated sentences—many noisy and irrelevant components were present in the raw text. These included inconsistencies such as punctuation anomalies, English numerals, and other non-Bangla elements that needed to be cleaned to ensure quality for downstream tasks like Named Entity Recognition (NER). The manual translation process, particularly for dialectal variations, introduced further irregularities. Since the ONUBAD dataset covers only three regions (Sylhet, Barishal, and Chittagong), we first collected standard Bangla versions of each sentence and then translated them into regional dialects, including Noakhali and Mymensingh through our hired annotators, specialized in regional dialects. These transformations sometimes resulted in mixed-language tokens, requiring a dedicated cleaning step. Some Initial data cleaning was manually done to get rid of the Punctuation anomaly and Mixed-language token, which has been shown in Table 2.

To apply further pre-processing, we developed an automated Python-based data cleaning and tokenization pipeline. This pre-processing step involved:

1. Extraneous symbols using Python's regular expressions (re) module.
2. Separating punctuation marks from words to ensure accurate tokenization.

English numerals were replaced with Bengali equivalents, which was done manually by the annotators. The tokenization was designed to prepare the data for annotation in a format suitable for NER tagging scheme.

The tokenization process, as described in Algorithm 1, is designed to convert raw Bangla Regional sentences into a structured, tokenized format suitable for downstream annotation and analysis. The algorithm begins by reading the input dataset, which is in CSV format, containing Bangla Regional. 5 different CSVs contains the Raw sentences of 5 different regions. Each sentence is processed one at a time. If the sentence is non-empty, it is passed through a regular expression-based tokenizer to split it into individual tokens. These tokens are typically Bangla Regional words, extracted by filtering out punctuation and whitespace. The regular expression ensures that unwanted characters such as punctuation marks and special symbols are excluded from the tokens. Each sentence is assigned a unique identifier (e.g., "Sentence: 1"), and the first token is associated with this identifier. The remaining tokens from the sentence are listed in subsequent rows, with their sentence identifier left as None (NaN), ensuring compatibility with common sequence labeling formats such as the CoNLL-style layout [62]. Finally, the tokenized output is written to a CSV file with two columns: one for the sentence identifier and the other for the individual words as shown in Table 3.

**Table 2**. **Sentence structure conversion by separating punctuation.**

| Conversion Type | Before | After |
|---|---|---|
| Punctuation Anomaly | তুই দুইঙ্গা ভাত হাইয়ো নে?? <br> (What did you eat rice with in the afternoon??) | তুই দুইঙ্গা ভাত হাইয়ো নে? <br> (What did you eat rice with in the afternoon?) |
| Mixed Language or Script | বর্তমান govt. অনেক প্রকল্প বাস্তবায়ন করেছে। <br> (The current government has implemented many projects.) | বর্তমান সরকার অনেক প্রকল্প বাস্তবায়ন করেছে। <br> (The current government has implemented many projects.) |

**Algorithm 1 Process Bangla regional sentences from data file.**

```
 1: Input: Read the data file (CSV) containing Bangla Regional sentences
 2: newRows ← [ ]
 3: sentenceID ← 1
 4: for each row in the data file do
 5:   sentence ← extract sentence from column
 6:   if sentence is not empty then
 7:     tokens ← split the sentence using regular expression
 8:     newRows ← newRows + ["Sentence: " + sentenceID, tokens[0]]
 9:     for i = 1 to length(tokens) - 1 do
10:       newRows ← newRows + [None, tokens[i]]
11:     end for
12:     sentenceID ← sentenceID + 1
13:   end if
14: end for
15: Output: Write newRows to CSV file with columns: Sentence #, Word
```

**Table 3**. Dataset structure for Sylhet region after pre-processing and tokenization phase (Followed for all 5 regions).

| Sentence # | Word (Sylhet Region) |
|---|---|
| 1 | ফুয়াটায় |
| NaN | সিলেট |
| NaN | থাকি |
| NaN | ঢাকাত |
| NaN | আইছে |

## Annotation scheme

Regarding the tagging scheme, various approaches such as BILUO, BIO, BIO2, IO, and BIOES have been proposed for tagging named entities in NER tasks [63]. The choice of tagging scheme is crucial, as it impacts the granularity and accuracy of the entity identification process. For languages like Bangla, which is a post-positional language, the BIO tagging scheme has been widely adopted by researchers [64]. Our proposed B-NER dataset follows this convention, with each entity chunk tagged as follows: the first token of an entity is labeled as "B-entity name" (Beginning), and subsequent tokens in the entity are labeled as "I-entity name" (Inside). All other tokens that do not belong to any named entity are tagged as "O-entity name" (Outside).

- B (Beginning): Beginning of a multi-word entity.
- I (Inside): Inside a multi-word entity.
- O (Outside): Outside any entity.

This approach is consistent with the BIO tagging scheme first introduced by Ramshaw and Marcus [65], ensuring that our dataset adheres to widely accepted conventions in the field of NER.

## Annotators identity

A total of ten annotators were recruited for the annotation task, with two annotators assigned to each of the five regional dialects: Chittagong, Sylhet, Barishal, Noakhali, and Mymensingh. The group consisted of both graduate and undergraduate students with academic backgrounds in Linguistics and Natural Language Processing (NLP). Their experience levels varied, ranging from 1 to 4 years, with an average age of 25.7 years. All annotators were native speakers of the

respective dialects and were selected to ensure balanced dialectal representation in the corpus. Prior to annotation, each annotator underwent a training session to familiarize themselves with the guidelines and the annotation interface. Detailed information of the annotators is presented in Table 4.

## Data annotation

For the annotators, we have provided detailed guidelines to ensure consistency and accuracy in tagging named entities. These guidelines help in categorizing and labeling various types of named entities within the text, following a standardized approach. The Location (LOC) tag refers to geographical entities, including cities, towns, landmarks, rivers, mountains, and other notable physical areas. The Person (PER) tag is applied to the names of individuals, covering public figures, common citizens, and fictional characters. The Organization (ORG) tag represents formal entities such as companies, educational institutions, government bodies, and NGOs. Food (FOOD) includes consumable items, ranging from raw ingredients to prepared dishes and beverages. The Animal (ANI) tag covers species and breeds within the biological kingdom Animalia, including both domesticated and wild animals. Color (COL) applies to terms that describe specific shades, hues, or composite colors. The Role (ROLE) tag is used for job titles or professional positions, including general roles like "teacher" or "manager," as well as more specific functional titles. Relationship (REL) encompasses familial and social connections, such as terms like "mother," "father," "friend," and "colleague." The Object (OBJ) tag refers to tangible, physical items, including everyday objects, tools, and machines. Finally, the Non-Entity (O) tag is used for words that do not belong to any named entity class, such as common nouns, verbs, and adjectives that are not associated with specific entities. Table 5, illustrates the annotation guidelines for our BIO scheme.

The Inter-Annotator Agreement (Cohen's Kappa) graph shown in Fig 3 evaluates the consistency and reliability of the annotations performed by multiple annotators. Cohen's Kappa [66] is a statistical measure that assesses the level of agreement between two annotators, taking into account the possibility of agreement occurring by chance. It is formally defined as:

$$\kappa = \frac{p_o - p_e}{1 - p_e}$$

where $p_o$ is the observed agreement proportion between annotators, and $p_e$ is the expected agreement by chance. In this study, each regional dialect was annotated by two annotators, and the Kappa scores were calculated for each pair. The graph visualizes these Kappa scores for each annotator, highlighting the level of agreement across different regional dialects. Higher Kappa scores, closer to 1, indicate strong agreement between annotators, which is critical for ensuring the quality and consistency of the annotated data. As seen in the graph, the agreement across most regions is high, demonstrating that the annotators were well-aligned in their understanding of the tagging guidelines. This agreement is essential for the reliability of the ANCHOLIK-NER dataset, as it ensures that the entity annotations accurately represent the dialectal variations.

**Table 4**. **Comprehensive overview of annotators' background and expertise.**

| Annotator | Region | Role | Age | Research Field | Experience |
|---|---|---|---|---|---|
| Annotator 1 | Chittagong | Graduate | 28 | Linguistics | 3 years |
| Annotator 2 | Chittagong | Under-graduate | 22 | NLP | 1 year |
| Annotator 3 | Sylhet | Graduate | 30 | NLP | 4 years |
| Annotator 4 | Sylhet | Under-graduate | 23 | Linguistics | 2 years |
| Annotator 5 | Barishal | Under-graduate | 24 | NLP | 1 year |
| Annotator 6 | Barishal | Under-graduate | 25 | Linguistics | 1 year |
| Annotator 7 | Khulna | Graduate | 27 | NLP | 3 years |
| Annotator 8 | Khulna | Under-graduate | 23 | Linguistics | 2 years |
| Annotator 9 | Mymensingh | Graduate | 29 | NLP | 4 years |
| Annotator 10 | Mymensingh | Under-graduate | 22 | Linguistics | 1 year |

**Table 5**. BIO Tagging scheme with examples for named entity recognition in Bangla regional dialects.

| Tag | Example | Explanation |
|---|---|---|
| B-LOC | লালবাগ কেল্লার (Lalbagh Fort) | Tags the starting word of a Location name |
| I-LOC | লালবাগ কেল্লার (Lalbagh Fort) | Tags the inside of a multi-word Location name |
| B-PER | লিওনেল মেসি (lionel Messi) | Tags the starting word of a Person name |
| I-PER | লিওনেল মেসি (lionel Messi) | Tags the inside of a multi-word Person name |
| B-ORG | পাবলিক বিশ্ববিদ্যালয়ে (Public University) | Tags the starting word of an Organization name |
| I-ORG | পাবলিক বিশ্ববিদ্যালয়ে (Public University) | Tags the inside of a multi-word Organization name |
| B-FOOD | মোরগের মাংস (Chicken Meat) | Tags the starting word of a Food name |
| I-FOOD | মোরগের মাংস (Chicken Meat) | Tags the inside of a multi-word Food name |
| B-ANI | মুরগির বাইচ্চা (Chicken Chick) | Tags the starting word of an Animal name |
| I-ANI | মুরগির বাইচ্চা (Chicken Chick) | Tags the inside of a multi-word Animal name |
| B-COL | কালা রং (Black Color) | Tags the starting word of a Color name |
| I-COL | কালা রং (Black Color) | Tags the inside of a multi-word Color name |
| B-ROLE | কামের মাইয়া (Domestic Worker) | Tags the starting word of a Role title |
| I-ROLE | কামের মাইয়া (Domestic Worker) | Tags the inside word of a multi-word Role title |
| B-REL | বড় আফার (Elder Sister) | Tags the starting word of a Relationship label |
| I-REL | বড় আফার (Elder Sister) | Tags the inside word of a multi-word Relationship label |
| B-OBJ | লাকড়ির চুলা (Wood Stove) | Tags the starting word of a Physical Object name |
| I-OBJ | লাকড়ির চুলা (Wood Stove) | Tags the inside word of a multi-word Physical Object name |
| O | দিনে, মুশকিল (Daytime, Difficult) | Tokens that do not belong to any named entity class |

The Tagging Speed (Time per 1000 Tokens) by Region graph shown in Fig 4 provides insights into the efficiency of the annotators when labeling entities in the dataset. The graph shows the average time each annotator took to tag 1000 tokens, grouped by region. This metric is important because it highlights the speed of annotation and reflects the annotators' familiarity with the linguistic features of their respective regional dialects. The results indicate that annotators from Mymensingh were the fastest averaging 6 min for the completion of 1000 tokens, completing the annotation in the least amount of time, while Chittagong and Sylhet annotators took slightly longer, around 9 and 9.5 mins repectively. The regions are color-coded, with Barishal taking an intermediate amount of time, followed by Khulna, which showed a similar efficiency. The graph also suggests that experience with the dialect and familiarity with the annotation process may have influenced the speed of tagging.

Following the annotation process, the dataset has been organized into separate sub-datasets for each regional dialect. The entire corpus is saved as CSV (comma-separated values) file, which is structured quite conveniently. A sample of the

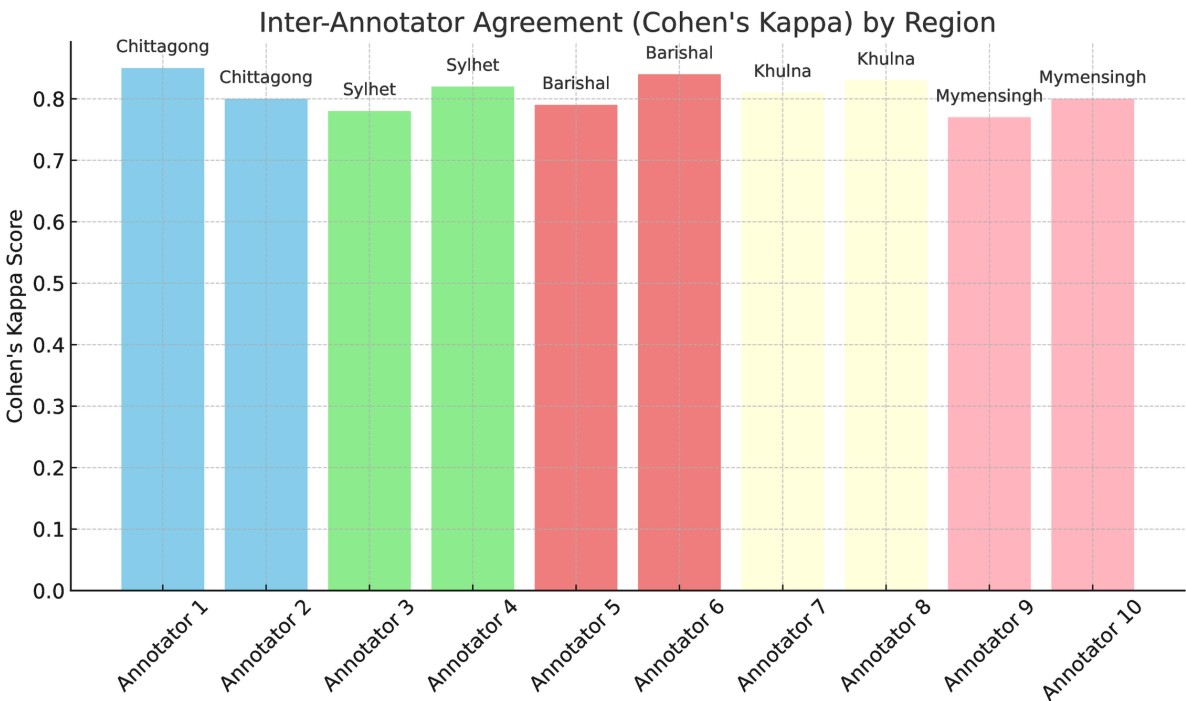

**Fig 3**. **Inter-annotator agreement (Cohen's Kappa) across different regions.**

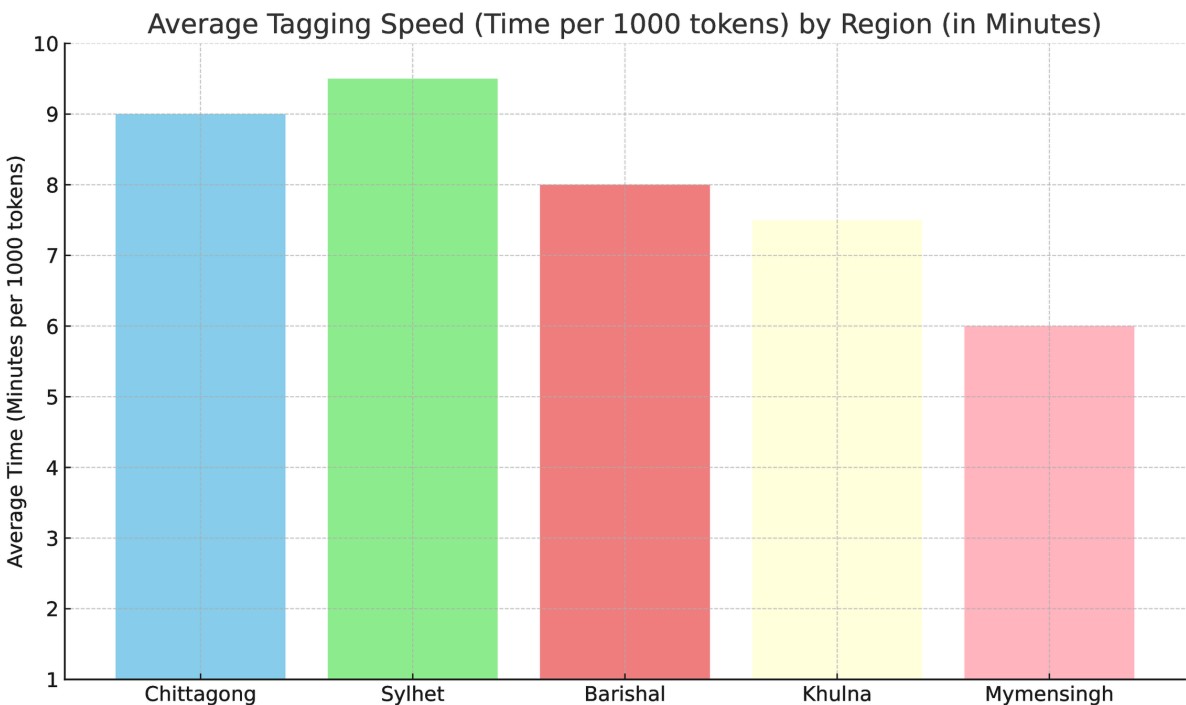

**Fig 4**. **Average tagging speed (time per 1000 tokens) by region in minutes.**

data structure is presented in Table 6, where the regions are displayed side by side for comparison purposes. Each entry in the dataset consists of:

1. Sentence Number – The sentence number in the ANCHOLIK-NER dataset for each regions.
2. Tokenized Words – Each word is treated as a separate token.
3. Named Entity Annotations – Assigned entity tags following the BIO scheme.

The Table 6 illustrates the variations in word structure across different regional dialects of Bangladesh for the same sentence, with specific attention to an important aspects. The presence of a "-" in the table indicates that there was no corresponding word in that dialect for the given word, highlighting a gap or difference in vocabulary between regions. For example, the word "সিলেট থাকি" in the Sylhet region doesn't have a match in the Chittagong region, which is represented by the "-" symbol. Two words had been merged into a single word "সিলেটভুন". This merging or splitting of words leads to discrepancies in word counts between the dialects, as seen in the varying number of words per sentence for each region. Additionally, it is important to note that there is no "-" in the actual CSV files; it has only been added in the table for comparison purposes. These differences highlight the linguistic diversity and the complex nature of regional dialects in Bangladesh.

## Outliers checking

Algorithm 2 was designed to ensure the quality and consistency of the annotations in the dataset. The algorithm checks for common issues in the BIO Tags column, such as outliers (tags that are incorrectly written in lowercase, like "o" instead of "O") and blank cells (missing annotations). The algorithm first processes the dataset by extracting all the unique tags and identifying any instances where tags deviate from the expected format. Specifically, it detects any outliers, such as lowercase tags or any missing tags, by checking each BIO tag against the standard format. Once any outliers or blank cells are identified, the algorithm prints the corresponding line numbers, making it easy to pinpoint where corrections are needed. The annotators are promptly notified of these discrepancies and instructed to review the flagged entries. By correcting these issues, annotators ensure that all tags are correctly formatted, and no tokens are left untagged.

## Dataset statistics

After the dataset was compiled, it was essential to conduct an analysis to identify both its strengths and limitations. The data analysis revealed crucial patterns and relationships within the dataset that were not initially apparent. Table 7 provides an overview of the dataset's content, showing that it contains 17,405 sentences, with sentence lengths ranging from 2 to 10 words. The dataset encompasses over 1 lac tokens. The analysis reveals that named entities account for only a small portion, with non-named entities making up approximately 83.4% of the dataset, while named entities comprise 16.6%.

Table 6. **Dataset consists of 3 columns for each region, with the first two generated by a Python script and the third (BIO-Tags) verified by Bangla Regional Language experts.**

| Sentence # | Sylhet | Chittagong | Barishal | Noakhali | Mymensingh | BIO Tags |
|---|---|---|---|---|---|---|
| 1 | ফুয়াটায় | ফোয়াইবা | পলাউগ্গা | পোলাডা | ছেড়াটা | O |
| NaN | সিলেট | সিলেটভুন | শ্যলেত | সিলেট | সিলেটতন | B-LOC |
| NaN | থাকি | - | দিয়া | থেইকা | - | O |
| NaN | ঢাকাত | ঢাহা | ধাহা | ঢাকায় | ঢাকাত | B-LOC |
| NaN | আইছে | আইস্যে | আইসেলে | আইসে | আইসে | O |

**Algorithm 2 Checking for unique tags and outliers in BIO tags.**

```
 1: Input: Read the data file (CSV) containing Bangla Regional sentences
 2: tags ← [ ] {List to store BIO Tags}
 3: uniqueTags ← [ ] {List to store unique BIO Tags}
 4: lineNo ← 0 {Line number tracker}
 5: for each line in the data file do
 6:   words ← extract words from the row (Sentence #, Word, BIO Tags)
 7:   wordList ← split the row into a list of words
 8:   if wordList is not empty then
 9:     lineNo ← lineNo + 1
10:     tags ← tags + [wordList[2]] {Add BIO Tag to tags list}
11:     if wordList[2] is lowercase or non-standard (e.g., "o" instead of "O") then
12:       PRINT "Outlier found at line: ", lineNo {Print line number for outlier}
13:     end if
14:     if wordList[2] is empty then
15:       PRINT "Blank tag found at line: ", lineNo {Print line number for blank tag}
16:     end if
17:   end if
18: end for
19: for each tag in tags do
20:   if tag NOT IN uniqueTags then
21:     uniqueTags ← uniqueTags + [tag] {Add unique tag to uniqueTags list}
22:   end if
23: end for
24: Output: Print the list of uniqueTags to identify all distinct BIO Tags
25: Output: Print all the line numbers where outliers and blank tags are found
```

**Table 7. Overview of our proposed dataset.**

| Dataset Attributes | Frequency |
|---|---|
| Total Number of sentences | 17.405 |
| Total Named Entities | 11,062 |
| Total Non-Named Entities | 90,755 |
| Sentence Length | [2-10] |
| Entities | 10 |
| Tagging Scheme | BIO |
| Number of Tags | 19 |

The word clouds presented in Figs 5, 6, 7, 8, and 9 provide a visual representation of the most frequent terms in each of the five regional dialects: Chittagong, Sylhet, Barishal, Noakhali, and Mymensingh. These clouds highlight the unique vocabulary and linguistic features characteristic of each region, with larger words indicating higher frequency.

Table 8 and Figs 10, 11, 12, 13, and 14 provide a detailed breakdown of the distribution and frequency of named entity types across the five regional dialects in the dataset. Table 8 lists the total instances of each named entity type for Barishal, Sylhet, Chittagong, Noakhali, and Mymensingh, while the accompanying figures visualize the proportional frequencies per region.

Across all dialects, Relation (REL) and Location (LOC) entities dominate the corpus, reflecting the conversational and place-centric nature of regional Bangla text. Conversely, classes such as Animal (ANI) and Object (OBJ) appear far less frequently, indicating a degree of class imbalance that may affect model generalization. The high proportion of Miscellaneous (O) tokens—over 80% of all entries—illustrates the natural sparsity of named entities relative to common tokens in real text. Minor variations in counts among regions (e.g., slightly higher LOC mentions in Chittagong and Sylhet) capture

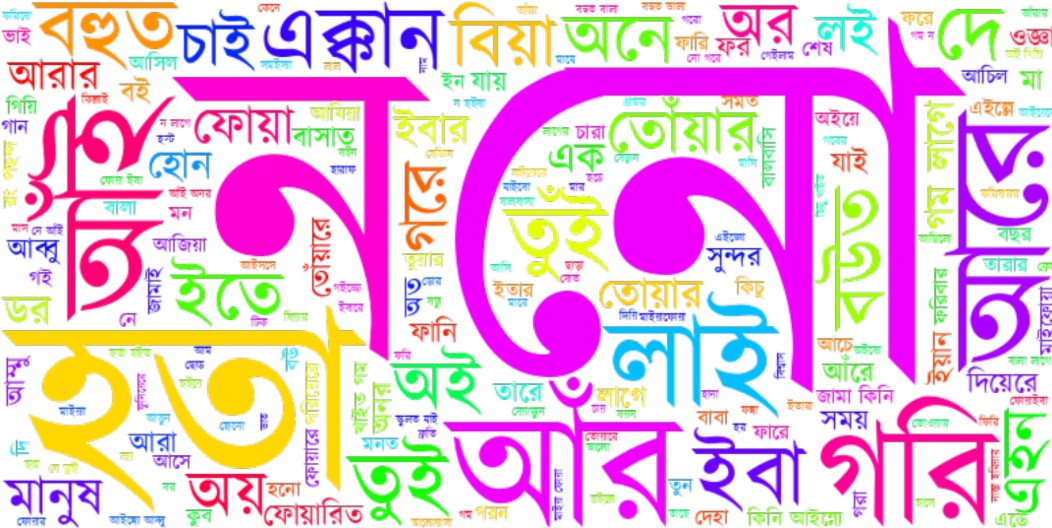

**Fig 5**. Chittagong.

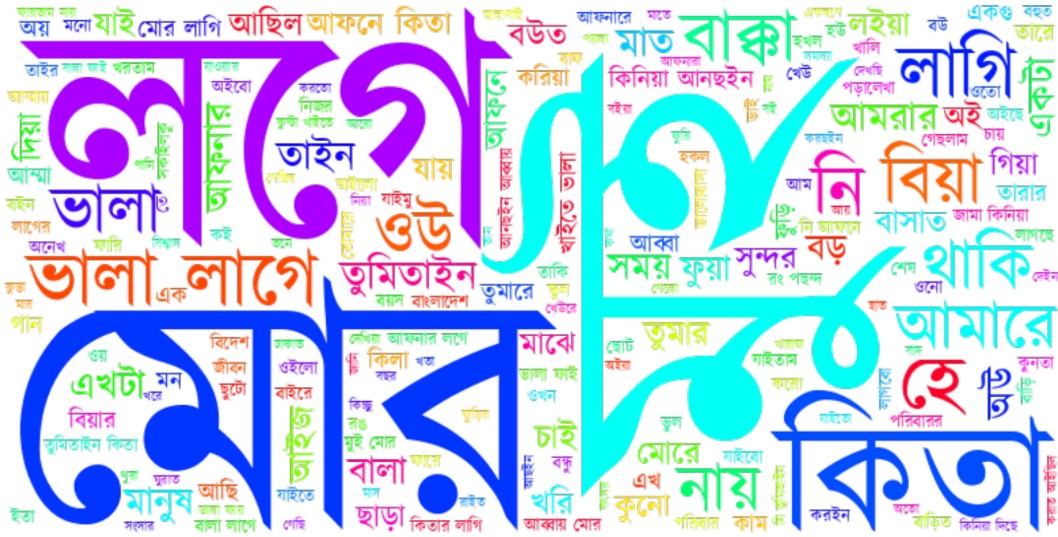

**Fig 6**. Sylhet.

the linguistic and topical diversity present across dialects. Collectively, these distributions confirm that the dataset captures both high-frequency general entities and lower-frequency, domain-specific ones, offering a realistic and challenging benchmark for NER in Bangla regional dialects.

## Data distribution

The dataset was split into 80% for training and 20% for testing, based on the total number of sentences. This distribution was done for each of the five regions. This 80:20 split ensures that the models are trained on a large portion of the data while being tested on a smaller, distinct subset for unbiased evaluation.

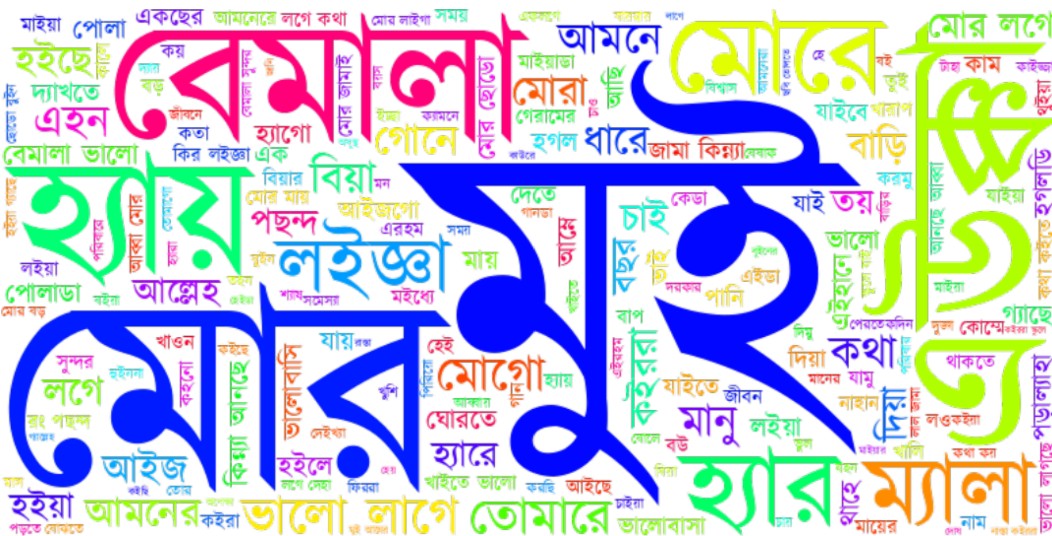

**Fig 7**. Barishal.

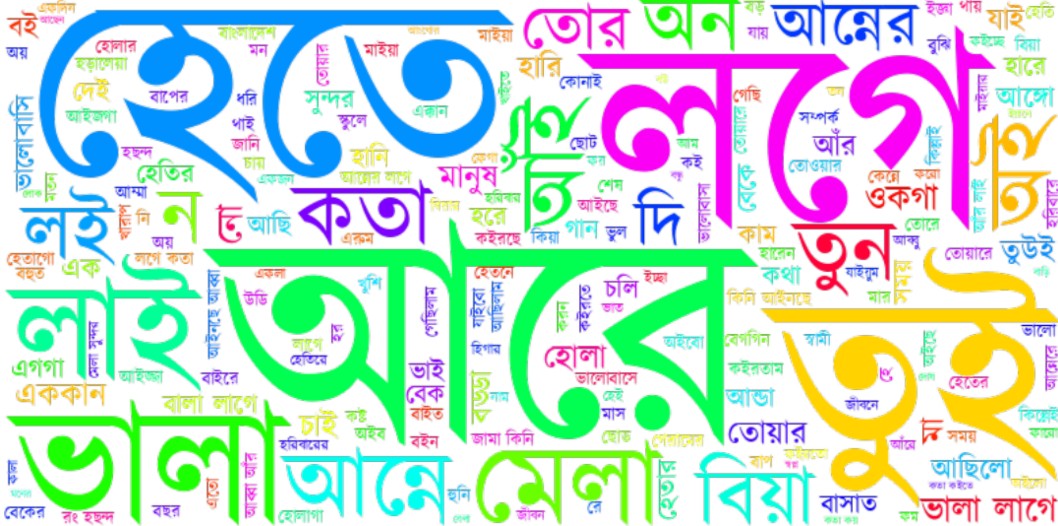

**Fig 8**. Noakhali.

## Methodology

In this section, we describe the methodology used to perform Named Entity Recognition (NER) for Bangla regional dialects. Following dataset preparation, preprocessing, and tokenization, we employ pretrained transformer-based models—Bangla BERT, Bangla Bert Base, and BERT Base Multilingual Cased—for fine-tuning on the ANCHOLIK-NER dataset. The objective is to evaluate the capability of existing architectures in handling dialectal variation rather than to develop new model architectures or training algorithms. To maintain a consistent and reproducible benchmarking setup, we intentionally use each model's default pretrained tokenizer and vocabulary without introducing dialect-specific

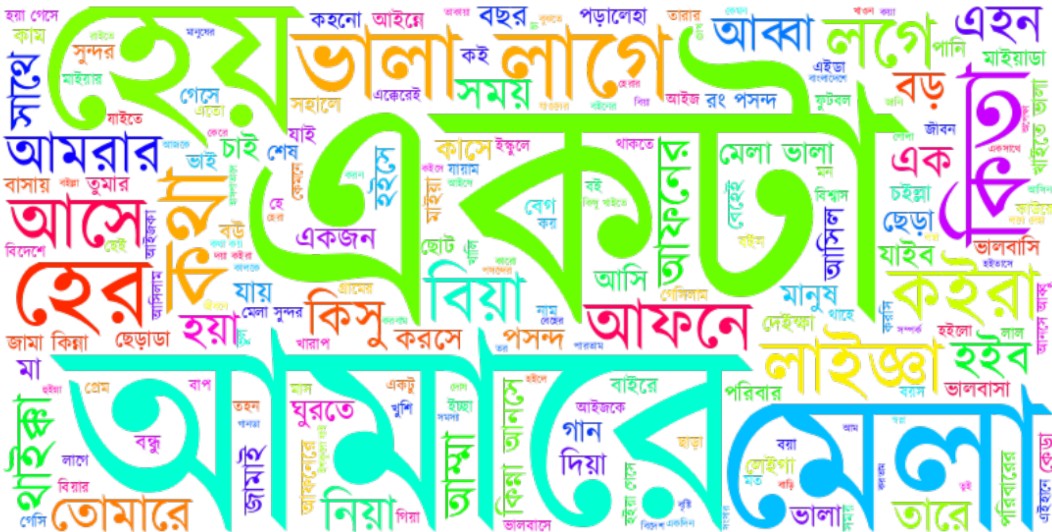

**Fig 9**. Mymensingh.

**Table 8**. Total instances of named entity types in five regions.

| Named Entity Type | Barishal | Sylhet | Chittagong | Noakhali | Mymensingh | Total Instances |
|---|---|---|---|---|---|---|
| Person (PER) | 39 | 38 | 39 | 39 | 39 | 194 |
| Location (LOC) | 369 | 371 | 377 | 361 | 362 | 1840 |
| Organization (ORG) | 139 | 141 | 139 | 141 | 140 | 700 |
| Food (FOOD) | 310 | 308 | 308 | 303 | 312 | 1541 |
| Animal (ANI) | 57 | 56 | 57 | 57 | 57 | 284 |
| Colour (COL) | 162 | 167 | 160 | 164 | 163 | 816 |
| Role (ROLE) | 114 | 107 | 109 | 111 | 113 | 554 |
| Relation (REL) | 681 | 677 | 676 | 676 | 676 | 3386 |
| Object (OBJ) | 352 | 348 | 348 | 350 | 349 | 1747 |
| Miscellaneous (O) | 17928 | 18750 | 18177 | 17957 | 17943 | 90,755 |

lexical adaptations or new subword tokens. This design choice allows us to isolate and assess the dataset-level challenges posed by regional Bangla dialects. Fig 15 gives an overview of our proposed methodology.

## Models overview

Named Entity Recognition (NER) in Bangla requires models that are specifically fine-tuned to understand the language's unique syntax, morphology, and regional variations. Given the complexity of Bangla and its regional dialects, transformer-based models like Bangla BERT, Bangla Bert Base, and BERT Base Multilingual Cased have shown great promise in achieving high accuracy for NER tasks.

**Bangla Bert.** Bangla BERT [67] is pre-trained specifically on Bangla data and fine-tuned for a range of Bangla NLP tasks, including NER. This model is particularly useful in NER tasks as it is fine-tuned to capture the syntactic and semantic intricacies of the Bangla language, ensuring high accuracy in identifying named entities. The choice of Bangla BERT ensures that the model is fine-tuned to the specific linguistic structure of Bangla, including its morphology and syntactic dependencies.

**Bangla Bert base.** The Bangla Bert Base [68] variant is based on the original BERT model architecture but pre-trained on a large corpus of Bangla text. It is well-suited for NER tasks due to its balanced structure, which optimizes

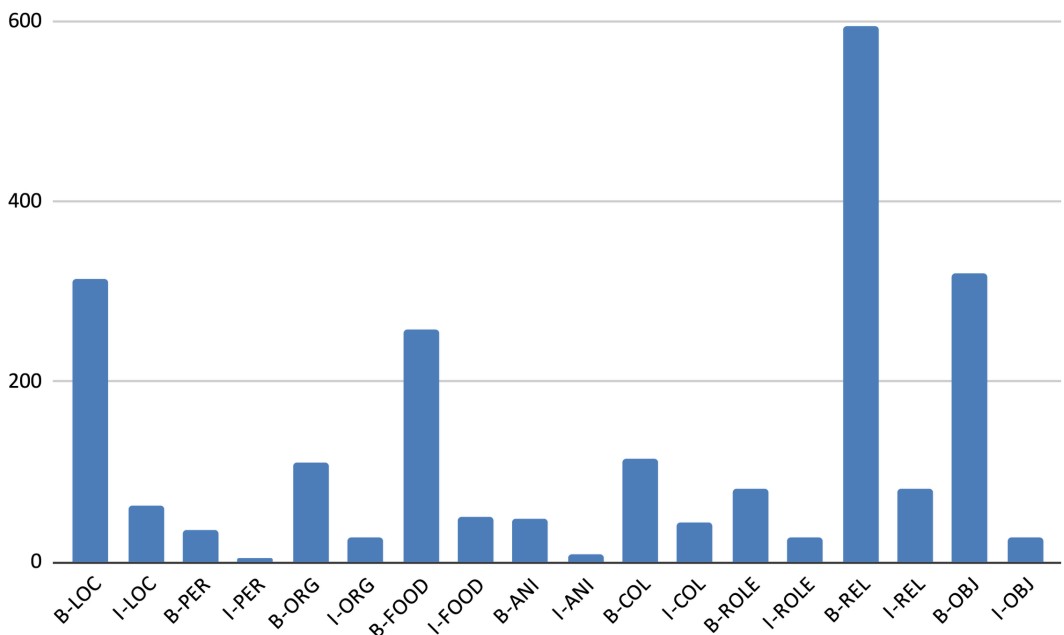

**Fig 10**. Frequency of named entities Chittagong dialects.

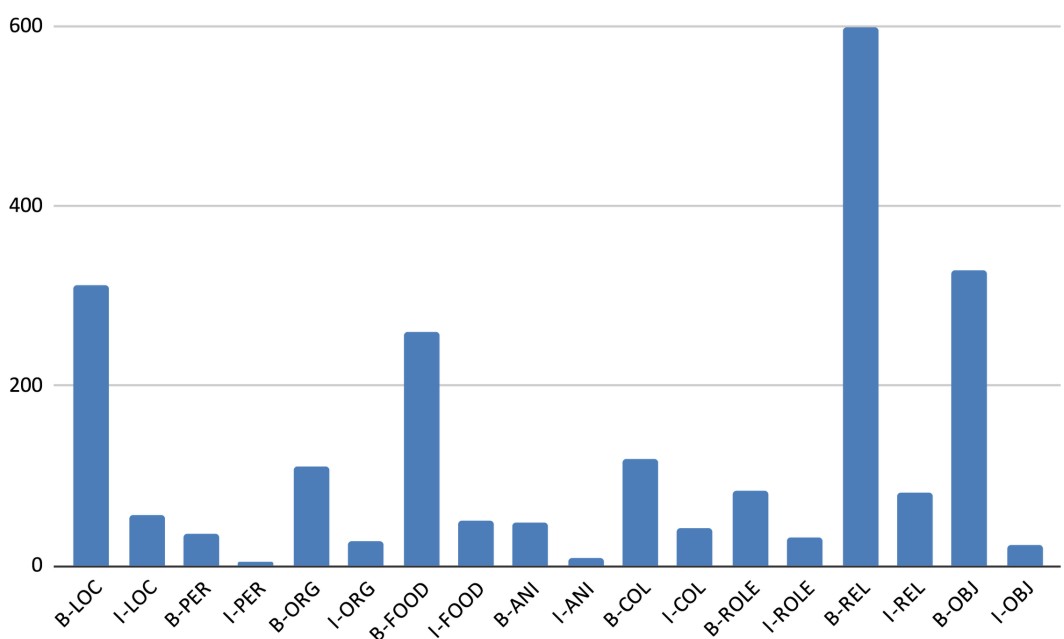

**Fig 11**. Frequency of named entities Barishal dialects.

performance while maintaining computational efficiency, especially when handling complex dialectal variations in regional Bangla. This variant is essential for situations where model performance needs to be optimized while managing computational resources. By using the BERT Base architecture, which has been proven to work effectively on various NLP tasks, we aim to achieve an optimal balance between computational cost and performance.

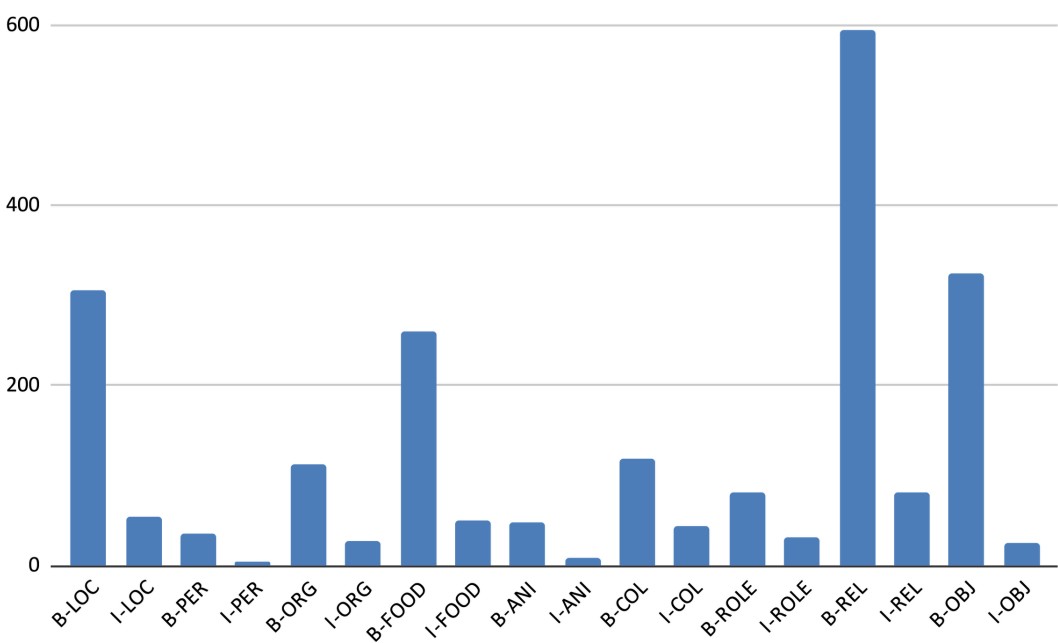

**Fig 12**. **Frequency of named entities Mymensingh dialects.**

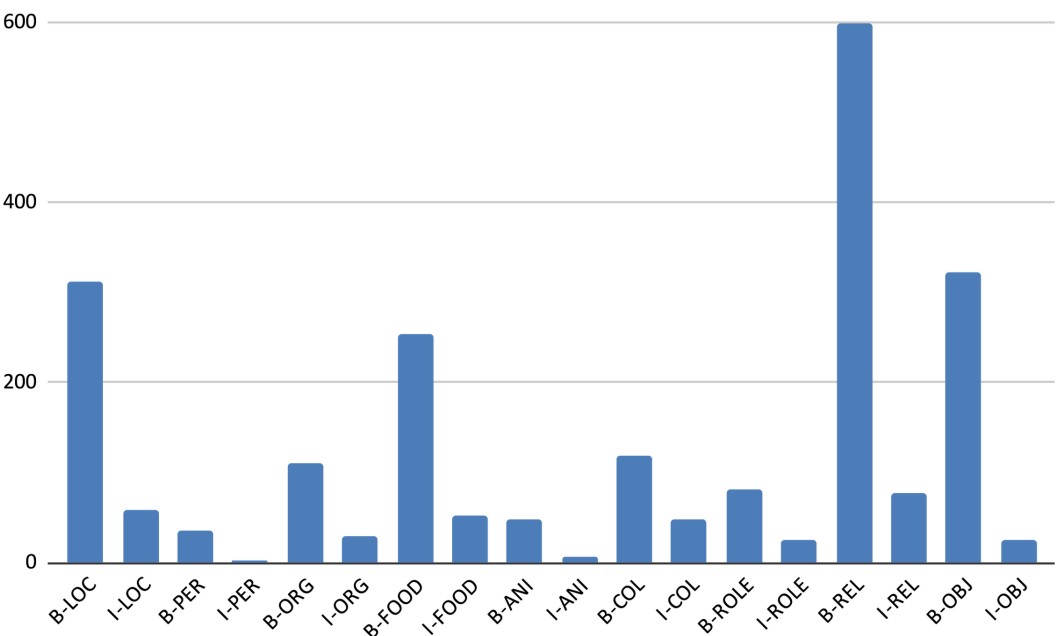

**Fig 13**. **Frequency of named entities Sylhet dialects.**

**BERT base multilingual cased.** The BERT Base Multilingual Cased model [69] is trained on a massive multilingual corpus, supporting over 100 languages, including Bangla. The model uses the transformer architecture to understand text in multiple languages simultaneously, making it highly useful for cross-linguistic tasks such as NER. This multilingual

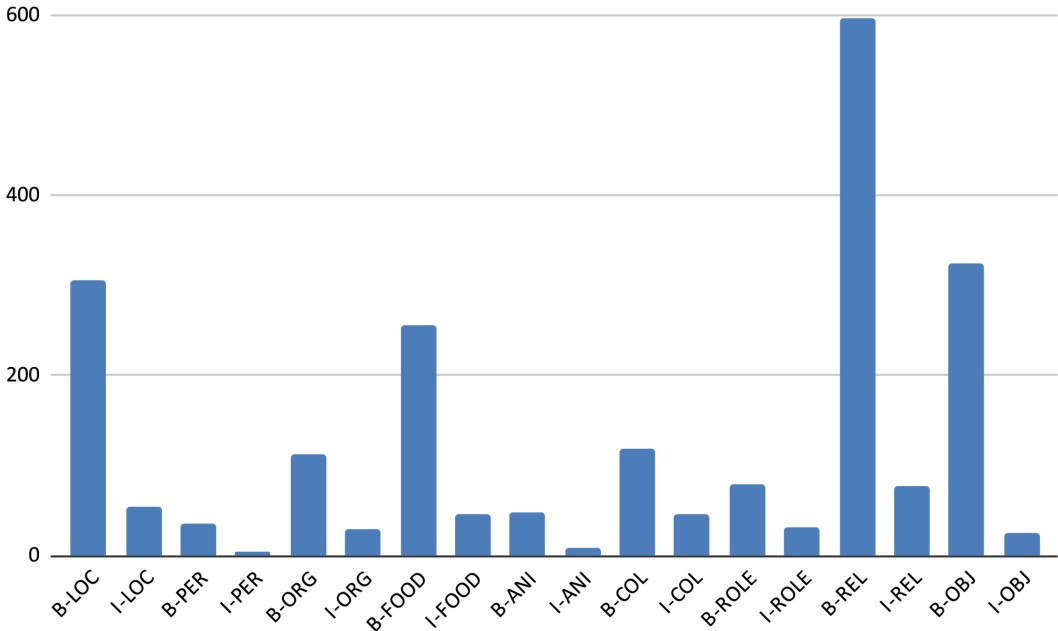

**Fig 14**. **Frequency of named entities Noakhali dialects.**

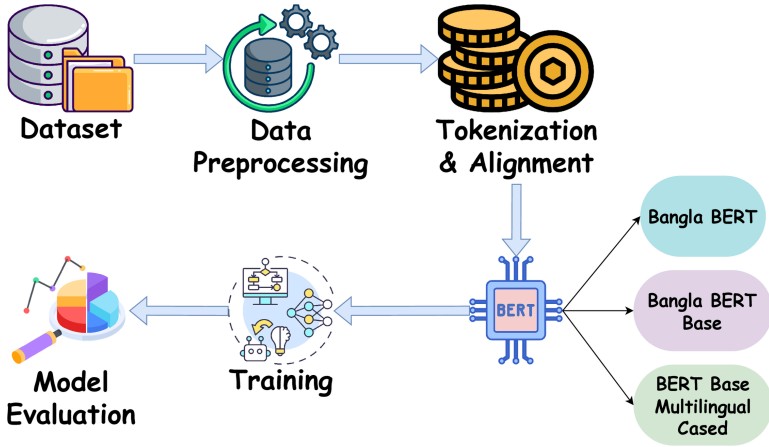

**Fig 15**. **Methodology.**

model was selected to evaluate how well a general-purpose language model can adapt to the specific requirements of Bangla regional dialects.

Although the model is not explicitly fine-tuned for intra-language dialectal variation of Bangla, we included it as a cross-lingual baseline to examine whether broad multilingual pretraining provides any transferable benefit for dialectal Bangla NER. This comparison offers valuable insights into the extent to which multilingual models capture dialectal diversity without targeted adaptation.

## Evaluation metrics

To assess the performance of the NER models, we use Precision, Recall, and F1-Score as our primary evaluation metrics. These metrics were selected because they provide a more balanced and informative evaluation of the model's performance, especially in tasks like NER, which often involve imbalanced datasets.

**Precision.** Precision measures the proportion of true positive named entities identified by the model out of all the named entities that the model identified as positive. A high precision score indicates that the model is making fewer false positive errors, meaning it correctly identifies relevant named entities without mistakenly classifying non-entities as entities. Precision is important when it is crucial to ensure that the entities detected by the model are relevant and correctly identified. Equation 1 displays the formula of precision.

$$\text{Precision} = \frac{TP}{TP + FP} \tag{1}$$

Where:

- $TP$ = True Positives
- $FP$ = False Positives

**Recall.** Recall measures the proportion of true positive named entities identified by the model out of all the actual positive named entities in the dataset. A high recall score indicates that the model is capturing most of the true named entities and not missing important information. This metric is crucial for tasks where it is important to minimize false negatives, especially when the named entities are important for the overall understanding of the text. Equation 2 displays the formula of recall.

$$\text{Recall} = \frac{TP}{TP + FN} \tag{2}$$

Where:

- $TP$ = True Positives
- $FN$ = False Negatives

**F1-Score.** The F1-Score is the harmonic mean of precision and recall, and it is particularly useful when dealing with imbalanced datasets. The F1-score provides a single metric that balances both precision and recall, ensuring that the model does not overemphasize one metric at the expense of the other. This metric is highly effective for NER tasks, where the correct identification of entities is critical, and both false positives and false negatives should be minimized [70]. In our NER tasks, especially for Bangla regional dialects, an imbalance in named entities (e.g., fewer occurrences of specific entity types) is common, making the F1-score an essential metric for evaluating the model's overall performance. By considering both precision and recall, the F1-score ensures that the model achieves a balance between identifying all relevant entities while avoiding the misclassification of non-entities as named entities. Equation 3 displays the formula for the F1-Score. Equation 3 displays the formula of F1-Score.

$$\text{F1-Score} = 2 \times \frac{\text{Precision} \times \text{Recall}}{\text{Precision} + \text{Recall}} \tag{3}$$

## Result analysis

The performance of three different BERT models—Bangla BERT, Bangla Bert Base, and BERT Base Multilingual Cased—was evaluated for Named Entity Recognition (NER) across five regional dialects of Bangla: Barishal, Chittagong, Mymensingh, Noakhali, and Sylhet. The models were trained with a particular learning rates (2e-5), different batch sizes (8, 16) and epochs(5, 10, 15, 20). Their performance was assessed based on precision, recall, and F1-score.

The Bangla BERT model (Table 9) was evaluated across five Bangla regional dialects with varying epochs and a learning rate with batch sizes of 8 and 16. The results show that Bangla BERT performed best in the Mymensingh region, achieving the highest F1-score of **82.268%** at **epoch 20**. In Barishal, it also performed well, reaching an F1-score of **81.481%** at **epoch 20**. Sylhet and Noakhali showed moderate performance, with Sylhet achieving a peak F1-score of **78.754%** at **epoch 20**, and Noakhali reaching **78.497%** at **epoch 20**. The Chittagong region, however, showed relatively

**Table 9**. **Performance of Bangla BERT.**

| Model | Batch Size | Region | Epoch | Precision | Recall | F1-score |
|---|---|---|---|---|---|---|
| Bangla BERT | 8 | Barishal | 05 | 0.69931 | 0.73465 | 0.71654 |
| | | | 10 | 0.77915 | 0.79602 | 0.78750 |
| | | | 15 | 0.80900 | 0.81046 | 0.80973 |
| | | | 20 | 0.83650 | 0.79422 | **0.81481** |
| | | Chittagong | 05 | 0.66775 | 0.65806 | 0.66287 |
| | | | 10 | 0.76334 | 0.69193 | 0.72588 |
| | | | 15 | 0.76329 | 0.71774 | 0.73981 |
| | | | 20 | 0.76627 | 0.74032 | **0.75307** |
| | | Mymensingh | 05 | 0.74353 | 0.76639 | 0.75479 |
| | | | 10 | 0.79446 | 0.82377 | 0.80885 |
| | | | 15 | 0.80360 | 0.82172 | 0.81256 |
| | | | 20 | 0.82780 | 0.81762 | **0.82268** |
| | | Noakhali | 05 | 0.65669 | 0.69459 | 0.67511 |
| | | | 10 | 0.78694 | 0.76350 | 0.77504 |
| | | | 15 | 0.77692 | 0.75232 | 0.76442 |
| | | | 20 | 0.79166 | 0.77839 | **0.78497** |
| | | Sylhet | 05 | 0.75412 | 0.74591 | 0.75000 |
| | | | 10 | 0.76363 | 0.76225 | 0.76294 |
| | | | 15 | 0.81106 | 0.77132 | **0.79069** |
| | | | 20 | 0.79482 | 0.78039 | 0.78754 |
| | 16 | Barishal | 05 | 0.66725 | 0.67689 | 0.67204 |
| | | | 10 | 0.72972 | 0.77978 | 0.75392 |
| | | | 15 | 0.77640 | 0.79602 | 0.78609 |
| | | | 20 | 0.78368 | 0.79783 | **0.79069** |
| | | Chittagong | 05 | 0.60202 | 0.57580 | 0.58862 |
| | | | 10 | 0.66944 | 0.64677 | 0.65791 |
| | | | 15 | 0.73445 | 0.70483 | 0.71934 |
| | | | 20 | 0.75043 | 0.69838 | **0.72347** |
| | | Mymensingh | 05 | 0.66358 | 0.73565 | 0.69776 |
| | | | 10 | 0.77354 | 0.79098 | 0.78216 |
| | | | 15 | 0.81069 | 0.80737 | 0.80903 |
| | | | 20 | 0.82244 | 0.82581 | **0.82413** |
| | | Noakhali | 05 | 0.56856 | 0.63314 | 0.59911 |
| | | | 10 | 0.72413 | 0.74301 | 0.73345 |
| | | | 15 | 0.77042 | 0.73743 | 0.75356 |
| | | | 20 | 0.77716 | 0.78584 | **0.78148** |
| | | Sylhet | 05 | 0.65277 | 0.68239 | 0.66725 |
| | | | 10 | 0.75591 | 0.75317 | 0.75454 |
| | | | 15 | 0.77614 | 0.76769 | 0.77189 |
| | | | 20 | 0.79777 | 0.78039 | **0.78899** |

lower performance compared to the other regions, with the highest F1-score of **75.307%** at **epoch 20**. Overall, Bangla BERT demonstrated strong performance, with its highest F1-scores observed in Mymensingh and Barishal.

The Bangla Bert Base model (Table 10) performed well across all regions, with the highest F1-score of **79.973%** in Barishal at **epoch 15** and **80.916%** in Mymensingh at **epoch 10**, demonstrating its ability to recognize named entities effectively. In Chittagong, the model achieved a peak F1-score of **71.469%** at **epoch 15**, while Sylhet saw its highest F1-score of **78.215%** at **epoch 15**. Noakhali exhibited a consistent improvement in performance, with a peak F1-score of **77.692%** at **epoch 20**. Overall, Bangla Bert Base showed solid performance across most regions, with particularly notable results in Mymensingh and Sylhet, while regions like Chittagong and Noakhali saw more modest improvements. These results suggest that the model performs well but could benefit from further fine-tuning for dialects with more complex linguistic features.

**Table 10**. **Performance of Bangla Bert base.**

| Model | Batch Size | Region | Epoch | Precision | Recall | F1-score |
|---|---|---|---|---|---|---|
| Bangla BERT Base | 08 | Barishal | 05 | 0.80711 | 0.76424 | 0.78509 |
| | | | 10 | 0.81292 | 0.76554 | 0.78852 |
| | | | 15 | 0.82216 | 0.77849 | **0.79973** |
| | | | 20 | 0.82753 | 0.77072 | 0.79812 |
| | | Chittagong | 05 | 0.76062 | 0.64314 | 0.69696 |
| | | | 10 | 0.75888 | 0.65379 | 0.70243 |
| | | | 15 | 0.77864 | 0.66045 | **0.71469** |
| | | | 20 | 0.78964 | 0.64980 | 0.71292 |
| | | Mymensingh | 05 | 0.79663 | 0.77645 | 0.78641 |
| | | | 10 | 0.82942 | 0.78986 | **0.80916** |
| | | | 15 | 0.81720 | 0.79284 | 0.80484 |
| | | | 20 | 0.82453 | 0.79135 | 0.80760 |
| | | Noakhali | 05 | 0.80613 | 0.73382 | 0.76828 |
| | | | 10 | 0.82013 | 0.73088 | 0.77293 |
| | | | 15 | 0.81892 | 0.73823 | 0.77648 |
| | | | 20 | 0.81451 | 0.74264 | **0.77692** |
| | | Sylhet | 05 | 0.81481 | 0.70375 | 0.75522 |
| | | | 10 | 0.80821 | 0.73852 | 0.77180 |
| | | | 15 | 0.84025 | 0.73157 | **0.78215** |
| | | | 20 | 0.82884 | 0.72739 | 0.77481 |
| | 16 | Barishal | 05 | 0.78035 | 0.74093 | 0.76013 |
| | | | 10 | 0.79782 | 0.76165 | 0.77932 |
| | | | 15 | 0.80636 | 0.75518 | **0.77993** |
| | | | 20 | 0.82419 | 0.75906 | 0.79028 |
| | | Chittagong | 05 | 0.74720 | 0.62183 | 0.67877 |
| | | | 10 | 0.76443 | 0.65246 | 0.70402 |
| | | | 15 | 0.77974 | 0.64580 | 0.70648 |
| | | | 20 | 0.78378 | 0.65645 | **0.71449** |
| | | Mymensingh | 05 | 0.79750 | 0.76304 | 0.77989 |
| | | | 10 | 0.81933 | 0.77049 | 0.79416 |
| | | | 15 | 0.81212 | 0.79880 | **0.80540** |
| | | | 20 | 0.83360 | 0.76900 | 0.80000 |
| | | Noakhali | 05 | 0.77777 | 0.71029 | 0.74250 |
| | | | 10 | 0.80913 | 0.72941 | 0.76720 |
| | | | 15 | 0.78615 | 0.75147 | 0.76842 |
| | | | 20 | 0.80868 | 0.73970 | **0.77265** |
| | | Sylhet | 05 | 0.79047 | 0.69262 | 0.73832 |
| | | | 10 | 0.82108 | 0.71488 | 0.76431 |
| | | | 15 | 0.82103 | 0.72739 | **0.77138** |
| | | | 20 | 0.82467 | 0.70653 | 0.76104 |

The BERT Base Multilingual Cased model (Table 11) showed strong performance across most regions, achieving the highest F1-scores in Mymensingh and Sylhet. In Mymensingh, it reached its peak F1-score of **82.611%** at **epoch 20**, while in Sylhet, it achieved **82.315%** at **epoch 20**, demonstrating its ability to effectively handle regional dialects. In Chittagong, the model outperformed the others, with a peak F1-score of **76.377%** at **epoch 20**, indicating its proficiency in recognizing named entities in this dialect. However, in Barishal and Noakhali, the performance was more modest, with Barishal achieving an F1-score of **77.863%** at **epoch 15** and Noakhali reaching **81.553%** at **epoch 20**. These results highlight the model's strength in multilingual contexts while also suggesting that further fine-tuning could improve it's performance in specific regional dialects.

**Table 11**. **Performance of BERT base multilingual cased.**

| Model | Batch Size | Region | Epoch | Precision | Recall | F1-score |
|---|---|---|---|---|---|---|
| BERT Base Multilingual Cased | 08 | Barishal | 05 | 0.72710 | 0.72448 | 0.72579 |
| | | | 10 | 0.77556 | 0.77416 | 0.77486 |
| | | | 15 | 0.78690 | 0.77055 | **0.77863** |
| | | | 20 | 0.78451 | 0.75971 | 0.77191 |
| | | Chittagong | 05 | 0.74456 | 0.70801 | 0.72582 |
| | | | 10 | 0.79777 | 0.74074 | 0.73820 |
| | | | 15 | 0.77390 | 0.74590 | 0.75964 |
| | | | 20 | 0.77531 | 0.75258 | **0.76377** |
| | | Mymensingh | 05 | 0.78199 | 0.78476 | 0.78337 |
| | | | 10 | 0.82411 | 0.80513 | 0.81451 |
| | | | 15 | 0.83455 | 0.80425 | 0.81912 |
| | | | 20 | 0.85682 | 0.79752 | **0.82611** |
| | | Noakhali | 05 | 0.73551 | 0.76977 | 0.75225 |
| | | | 10 | 0.80344 | 0.77943 | 0.79125 |
| | | | 15 | 0.79355 | 0.80052 | 0.79702 |
| | | | 20 | 0.81914 | 0.81195 | **0.81553** |
| | | Sylhet | 05 | 0.74363 | 0.73100 | 0.73726 |
| | | | 10 | 0.80471 | 0.76228 | 0.78292 |
| | | | 15 | 0.84789 | 0.77211 | 0.80823 |
| | | | 20 | 0.82053 | 0.82578 | **0.82315** |
| | 16 | Barishal | 05 | 0.70953 | 0.73261 | 0.72088 |
| | | | 10 | 0.75559 | 0.73170 | 0.74346 |
| | | | 15 | 0.77674 | 0.75429 | 0.76535 |
| | | | 20 | 0.79061 | 0.76061 | **0.77532** |
| | | Chittagong | 05 | 0.69299 | 0.66494 | 0.67868 |
| | | | 10 | 0.75837 | 0.72179 | 0.73962 |
| | | | 15 | 0.78317 | 0.72179 | 0.75123 |
| | | | 20 | 0.79329 | 0.73385 | **0.76241** |
| | | Mymensingh | 05 | 0.74825 | 0.75819 | 0.75318 |
| | | | 10 | 0.79464 | 0.78830 | 0.79146 |
| | | | 15 | 0.82384 | 0.80779 | **0.81574** |
| | | | 20 | 0.81478 | 0.81045 | 0.81261 |
| | | Noakhali | 05 | 0.70320 | 0.73286 | 0.71772 |
| | | | 10 | 0.78892 | 0.80140 | 0.79511 |
| | | | 15 | 0.78747 | 0.78471 | 0.78609 |
| | | | 20 | 0.80926 | 0.78295 | **0.79589** |
| | | Sylhet | 05 | 0.68643 | 0.68275 | 0.68458 |
| | | | 10 | 0.78133 | 0.76318 | 0.77215 |
| | | | 15 | 0.80461 | 0.74709 | 0.77479 |
| | | | 20 | 0.79390 | 0.76764 | **0.78055** |

## Weighted-loss fine-tuning on Bangla BERT

ANCHOLIK-NER exhibits a substantial imbalance in entity distribution across all five dialect regions, as shown in Table 8. High-frequency categories such as `Miscellaneous`, `REL`, `LOC`, and `FOOD` dominate the corpus, each appearing thousands of times, whereas low-frequency categories such as `PER` (194 instances), `ANI` (284), `ROLE` (554), and `ORG` (700) occur comparatively rarely. This imbalance implies that a standard cross-entropy objective, which treats all classes uniformly, inherently biases the model toward majority labels while providing insufficient gradient signal for minority entity types. As a result, low-frequency classes are more susceptible to misclassification, a trend that is also visible in the dialect-wise confusion matrices (Figs 16, 17, 18, 19, 20).

To mitigate this skew and to better understand the extent to which class imbalance affects model behavior, we introduced a class-weighted loss for Bangla BERT. Each BIO-tag label was assigned a weight inversely proportional to its frequency in the training corpus, ensuring that rarer entity types receive proportionally higher emphasis during optimization. The weighted loss is defined as:

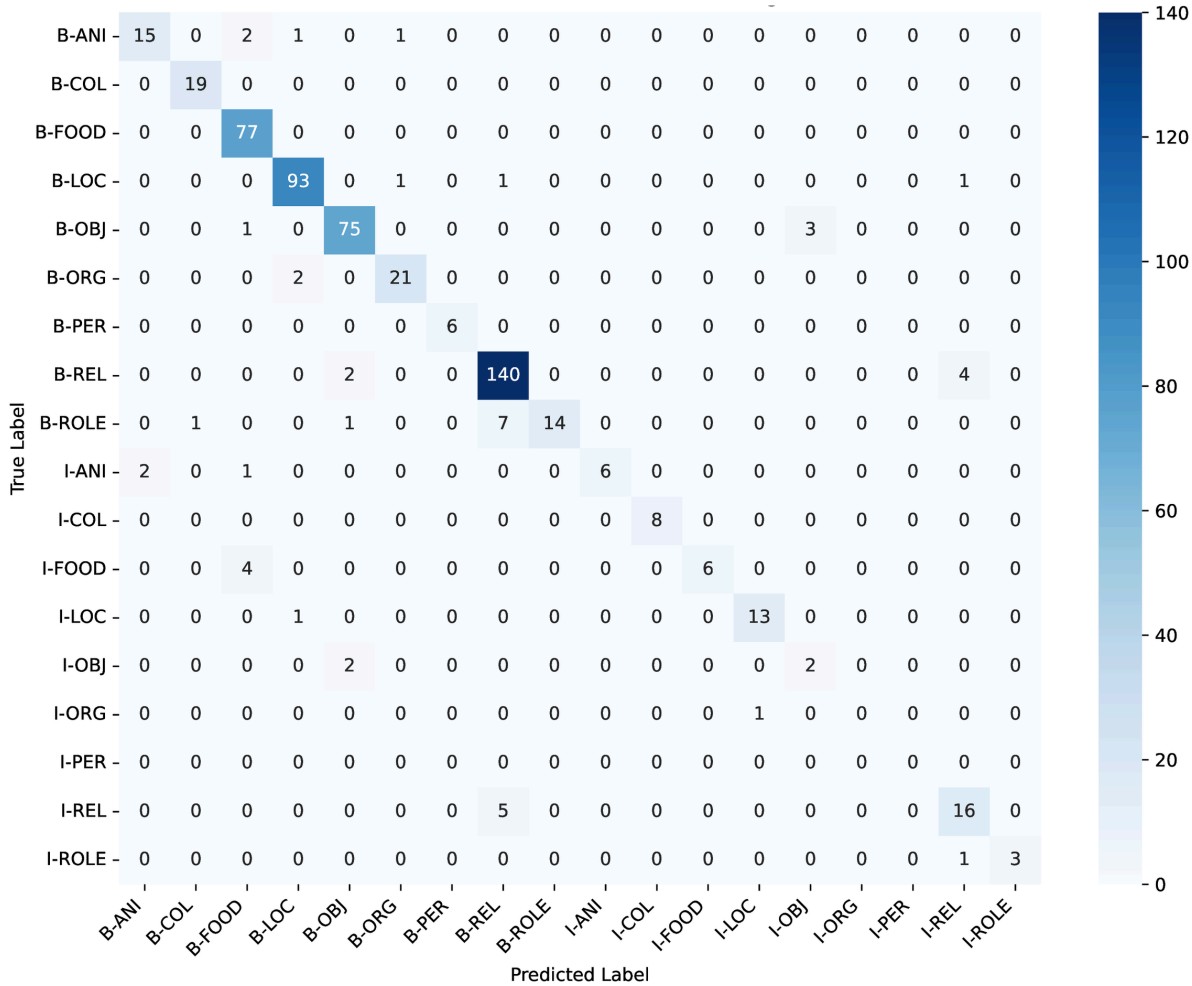

**Fig 16**. Confusion matrices for the best performing model across Barishal regional dialect.

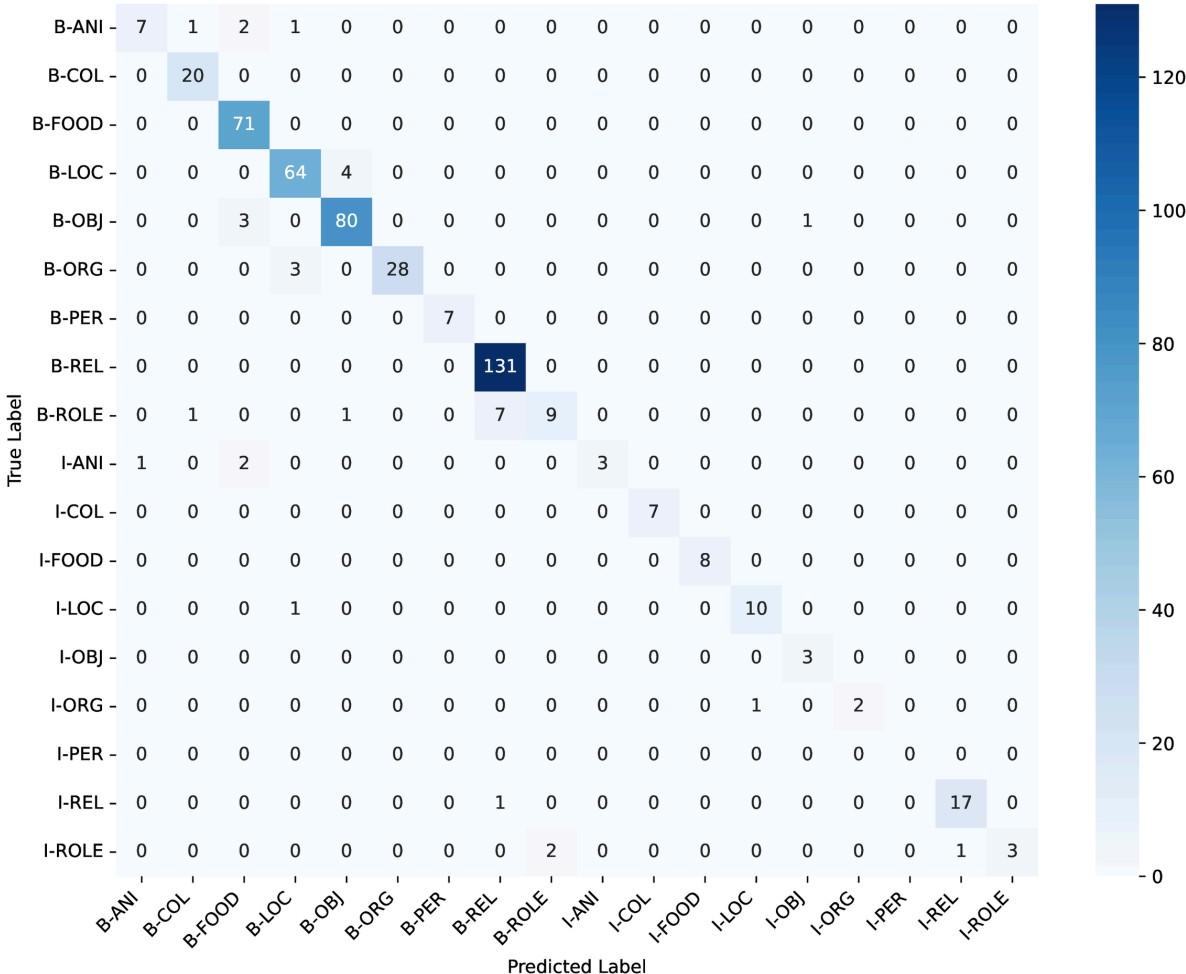

**Fig 17. Confusion matrices for the best performing model across Mymensingh regional dialect.**

$$\mathcal{L}_{\text{weighted}} = -\sum_{i=1}^{N} \sum_{c=1}^{C} w_c \, y_{i,c} \log p_{i,c}, \tag{4}$$

where $N$ is the number of tokens, $C$ is the number of BIO-tag classes, $y_{i,c}$ is the one-hot ground truth label for token $i$, $p_{i,c}$ is the predicted probability for class $c$, and $w_c$ is the weight assigned to class $c$. The class weights are computed as:

$$w_c = \frac{1}{f_c + \epsilon}, \tag{5}$$

with $f_c$ representing the empirical frequency of class $c$ in the training data and $\epsilon$ a small constant added for numerical stability. By assigning larger weights to rare labels, this loss formulation encourages the model to attend more closely to minority entity types without modifying the underlying model architecture or training pipeline. This weighted-loss experiment provides a controlled analysis of imbalance sensitivity and complements the baseline results by revealing performance gains for minority NER categories.

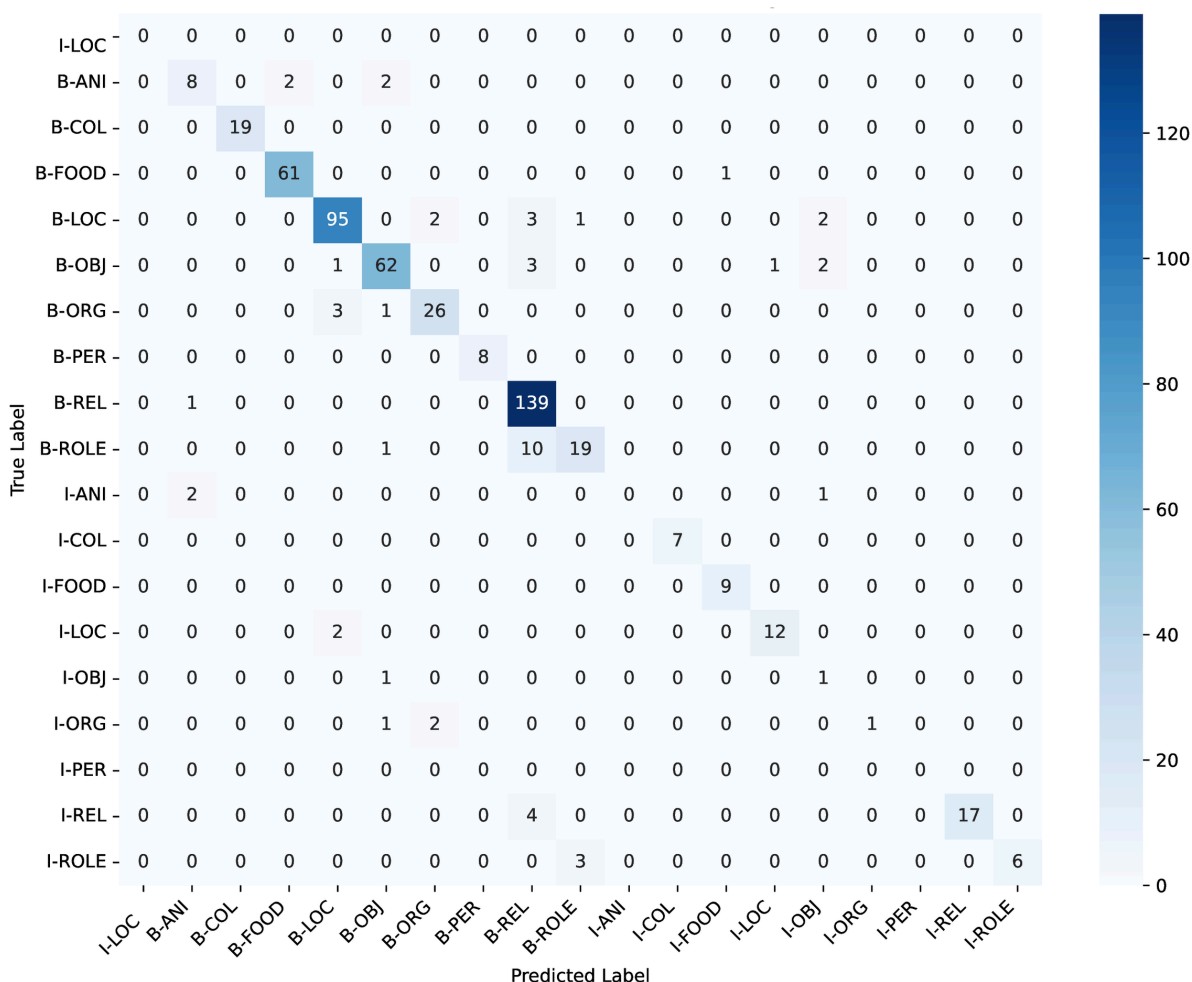

**Fig 18**. Confusion matrices for the best performing model across Chittagong regional dialect.

As shown in Table 12, the entity-wise F1-scores under the weighted-loss setting vary substantially across dialect regions, revealing which entity types remain challenging for the model. High-frequency categories such as COL, FOOD, LOC, and REL achieve relatively strong and stable performance across all regions, reflecting their consistent lexical patterns and richer training signals. In contrast, low-frequency or semantically variable categories such as ANI and ROLE show noticeably lower F1-scores in several dialects—particularly in Chittagong and Noakhali—indicating persistent difficulty despite the weighting strategy.

Certain minority categories, such as PER and ORG, achieve comparatively higher F1-scores in multiple regions (e.g., PER ranging from 0.80 to 0.90), but their performance remains inconsistent across dialects, suggesting dialect-specific variability in surface forms and contextual usage. Overall, the results highlight that entity difficulty is strongly correlated with both frequency and dialectal variation, and that some low-resource entity types continue to pose challenges for the model even under a weighted-loss regime.

The cross-dialect variation observed in Table 12 and for the Bangla BERT also relates to the degree of lexical overlap between individual dialects and Standard Bangla. Dialects such as Barishal and Mymensingh show higher overall F1-scores across several entity types, which may be attributed to their closer lexical proximity to Standard Bangla in

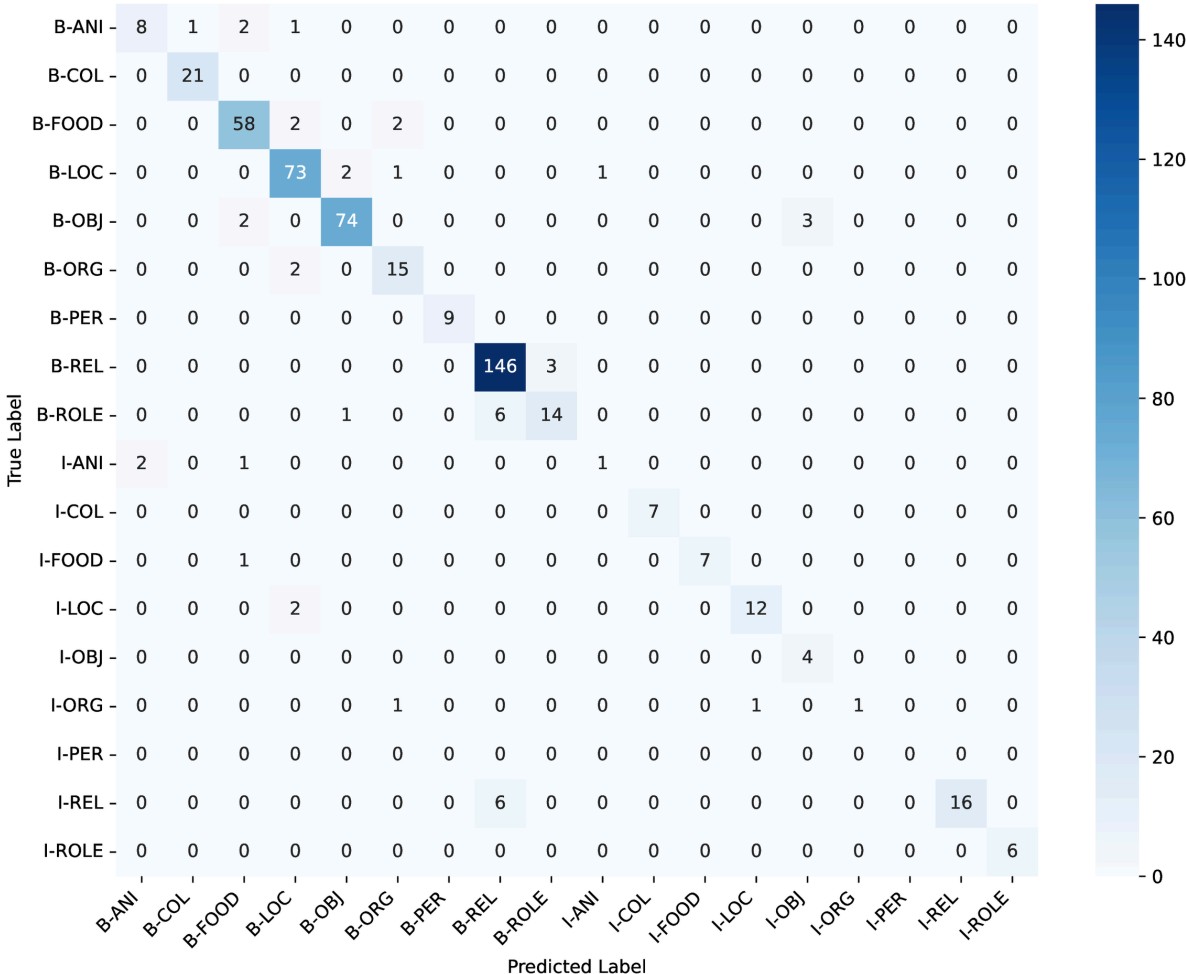

**Fig 19. Confusion matrices for the best performing model across Noakhali regional dialect.**

both vocabulary and morphological structure. In contrast, the Chittagong and Noakhali dialects exhibit more pronounced deviations in surface forms, including phonological shifts, unique lexical items, and dialect-specific word segmentation patterns. These characteristics reduce token overlap with Standard Bangla and hinder the effectiveness of pretrained language models whose vocabularies and subword units are primarily optimized for the standard form. As a result, categories with high morphological variability or limited training instances—such as ANI, ROLE, and ORG—tend to show larger performance drops in these dialects.

## Error analysis and per-entity difficulty

To better understand model behavior across dialects, we conduct an error analysis focusing on per-entity performance and dialect-specific variation. The entity-wise F1-scores presented in Table 12 show a clear separation between high-frequency and low-frequency categories. Frequent and lexically stable entity types such as COL, FOOD, LOC, and REL achieve consistently high performance across all five dialect regions, reflecting their richer contextual representation in the training data. In contrast, low-frequency or semantically diffuse categories, including ANI, ROLE, and ORG, remain

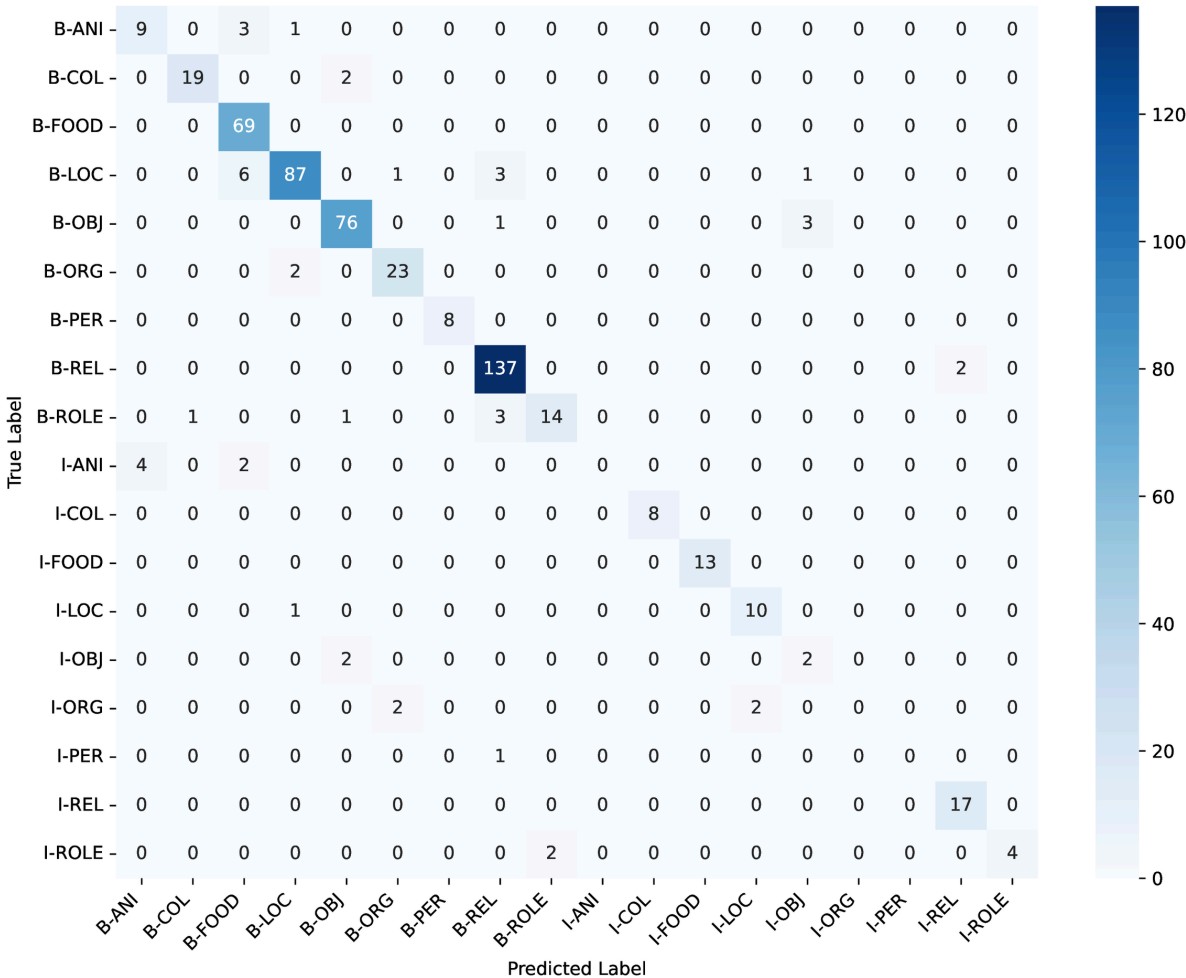

**Fig 20. Confusion matrices for the best performing model across Sylhet regional dialect.**

substantially more challenging. These classes often exhibit greater morphological variation and appear in less predictable syntactic contexts, which contributes to their lower recall and increased confusion with high-frequency classes.

The dialect-wise confusion matrices (Figs 16, 17, 18, 19, 20) further reveal systematic patterns of misclassification. Minority labels such as ANI and ROLE are frequently misassigned to dominant categories such as REL and LOC, indicating that the model defaults to high-frequency entity types when contextual cues are weak or when dialect-specific lexical forms diverge from Standard Bangla. These patterns are most prominent in the Chittagong and Noakhali dialects, where phonological shifts, unique vocabulary, and reduced token overlap with Standard Bangla make boundary detection more difficult. In contrast, dialects such as Barishal and Sylhet show comparatively higher per-entity performance, likely due to their closer lexical and morphological proximity to the standard language.

Across the confusion matrices, several recurrent failure modes emerge. First, the model frequently confuses ROLE with REL and ORG with LOC, reflecting semantic proximity and overlapping lexical cues across dialects. Second, although less dominant, there is observable confusion between FOOD and OBJ in multiple regions, indicating that certain nouns referring to consumable items or everyday objects are not consistently distinguished by the model. Third, the ANI category shows the highest variability and is often scattered across COL, FOOD, and LOC, especially in Chittagong and Noakhali, where

**Table 12**. Entity-wise F1-scores across the five dialect regions for the weighted-loss on Bangla BERT model.

| Entity | Barishal | Chittagong | Mymensingh | Noakhali | Sylhet |
|---|---|---|---|---|---|
| ANI | 0.7568 | 0.4375 | 0.6364 | 0.4800 | 0.4667 |
| COL | 0.9744 | 0.9744 | 0.8636 | 0.9545 | 0.8837 |
| FOOD | 0.8941 | 0.7821 | 0.9221 | 0.7534 | 0.8466 |
| LOC | 0.8824 | 0.7331 | 0.7742 | 0.7725 | 0.8075 |
| OBJ | 0.7327 | 0.6392 | 0.8421 | 0.7644 | 0.7895 |
| ORG | 0.7692 | 0.7143 | 0.8254 | 0.5778 | 0.7719 |
| PER | 0.8000 | 0.8889 | 0.8750 | 0.9000 | 0.8235 |
| REL | 0.7557 | 0.7667 | 0.8754 | 0.8397 | 0.8218 |
| ROLE | 0.7368 | 0.6429 | 0.5625 | 0.6222 | 0.7179 |

dialect-specific terms diverge the most from Standard Bangla. These error patterns highlight the systematic directions in which misclassification occurs and provide clear insight into how entity semantics, class imbalance, and dialectal lexical variation jointly shape model behaviour.

Overall, this analysis demonstrates that entity-level difficulty is determined by both structural factors (such as class imbalance) and linguistic factors (such as dialectal divergence). High-resource entity types benefit from consistent distribution and shared lexical patterns across dialects, whereas low-resource entities—particularly those with dialect-specific surface forms—remain challenging even under weighted-loss training. These findings underscore the need for future dialect-aware pretraining and more balanced data augmentation strategies to improve robustness across all entity categories.

In conclusion, the performance analysis of the three models—Bangla BERT, Bangla BERT Base, and BERT Base Multilingual Cased—shows that while BERT Base Multilingual Cased delivers the strongest overall performance across most regions, Bangla BERT also performs competitively in several cases. Mymensingh achieved its highest F1-score of **82.611%** with BERT Base Multilingual Cased at **epoch 20**, and Sylhet similarly reached **82.315%**. Barishal, however, obtained its best performance with Bangla BERT, achieving an F1-score of **81.481%** at **epoch 20**. Chittagong and Noakhali displayed comparatively lower results, with Chittagong reaching a peak F1-score of **75.307%** using Bangla BERT and Noakhali achieving **81.553%** with BERT Base Multilingual Cased, both at **epoch 20**. The confusion matrices across all regions reveal clear patterns behind these differences: high-frequency categories such as FOOD, LOC, and REL are consistently well recognized, whereas low-resource and semantically variable classes—particularly ANI, ROLE, and ORG—remain the most challenging. Common failure modes include frequent misclassification of ROLE as REL, ORG collapsing into LOC, and occasional confusion between FOOD and OBJ, alongside scattered predictions of ANI into multiple dominant categories. These errors are especially pronounced in the Chittagong and Noakhali dialects, where greater lexical divergence from Standard Bangla reduces token overlap and increases ambiguity. Overall, the findings indicate that while multilingual models generalize well across dialects, dialect-aware fine-tuning and improved representation of minority entity types are crucial for addressing systematic errors and improving robustness in linguistically diverse regions.

## Discussion

The findings of this study provide valuable insights into the challenges of Named Entity Recognition (NER) across Bangla regional dialects and the adaptability of transformer-based language models. Although multilingual BERT (mBERT) is not explicitly trained for intra-language dialectal variation, its inclusion as a cross-lingual baseline offers important evidence on the extent to which multilingual pretraining can transfer knowledge across dialects of the same language. The model achieved competitive performance in some regions, yet its lower scores in dialects such as Chittagong highlight the need for dialect-specific fine-tuning and corpus adaptation. In contrast, Bangla BERT consistently demonstrated

stronger results, suggesting that language-specific pretraining better captures the morphological and lexical nuances of Bangla. Beyond NER, these observations emphasize the broader significance of dialectal resources like ANCHOLIK-NER in advancing inclusive, dialect-aware NLP and facilitating the development of large language models (LLMs) that more accurately represent the linguistic diversity of Bangla.

## Conclusion and future work

In this study, we introduced ANCHOLIK-NER, the first benchmark dataset for Named Entity Recognition (NER) in Bangla regional dialects, specifically focusing on Sylhet, Chittagong, Barishal, Noakhali, and Mymensingh. The dataset was developed to bridge the gap in NER resources for Bangla dialects, which have been underrepresented in computational linguistics. Our results demonstrate that the Bangla BERT model, fine-tuned specifically for Bangla, outperforms the other models, including Bangla Bert Base and BERT Base Multilingual Cased, in recognizing named entities across all regions, particularly in Mymensingh and Barishal. Despite the strong overall performance, regions like Chittagong and Noakhali presented challenges, with relatively lower precision and recall scores, suggesting the need for further fine-tuning or additional data for these regions. The proposed ANCHOLIK-NER dataset provides a valuable resource for training and evaluating NER models tailored to Bangla regional dialects, and it can contribute to the development of more inclusive and dialect-aware NLP systems. The findings from this study underline the importance of using region-specific models to improve NER performance and highlight the potential for further research in dialectal variations within the Bangla language.

While ANCHOLIK-NER fills a significant gap in NER resources for Bangla regional dialects, several challenges remain. First, the dataset currently covers only five regions, leaving many other dialects and sub-dialects underrepresented. Further expansion of the dataset to include more diverse regions and their specific linguistic features would enhance the generalizability of the models. Additionally, though the current study uses existing transformer models, future research could explore the potential of more advanced techniques, such as incorporating hybrid models or leveraging unsupervised learning approaches to capture dialectal variations more effectively. Moreover, despite the strong results for most regions, Chittagong and Noakhali showed relatively lower performance, indicating that more refined techniques, such as domain adaptation or regional-specific data augmentation, are necessary to address these specific dialects' challenges.

Future work will focus on enhancing the performance of the models in regions like Chittagong and Noakhali, where the models showed lower accuracy. This can be achieved through additional training data, further fine-tuning, or the incorporation of region-specific linguistic features. Additionally, exploring more advanced pre-training techniques or leveraging other language models for Bangla, such as multilingual transformers with fine-tuning on dialect-specific corpora, could improve the overall model performance. Furthermore, expanding the dataset to include more dialects and regional variations could help in building a more robust NER system that performs well across all Bangla dialects.

Despite the effectiveness of the baseline transformer models used in this study, an important limitation is that we did not adapt or extend the models to handle dialect-specific lexical variations such as merged, missing, or regionally altered word forms. These dialectal features often lead to fragmented subword tokenization in standard Bangla or multilingual BERT tokenizers, which may contribute to lower performance in dialects with stronger orthographic or morphological divergence, such as Chittagong and Noakhali. Future work will explore dialect-aware modeling strategies, including vocabulary extension, tokenizer adaptation, and lightweight adapter- or LoRA-based fine-tuning to better capture region-specific lexical patterns.

Beyond the immediate scope of NER, the ANCHOLIK-NER dataset also provides a foundation for future exploration with large language models (LLMs). While the present study focused on fine-tuning discriminative transformer models to ensure controlled, reproducible benchmarking, generative LLMs such as LLaMA or Mistral represent a promising direction for future research. These models can potentially capture deeper dialectal nuances and socio-linguistic patterns through instruction-tuning or zero-shot evaluation. Incorporating ANCHOLIK-NER into such evaluations may offer insights into

how well current LLMs generalize to low-resource regional varieties of Bangla and guide the development of more dialect-aware generative systems. As the dataset expands and more region-specific corpora become available, ANCHOLIK-NER can serve as a valuable resource for building inclusive, linguistically diverse LLM-based applications.

## Author contributions

**Conceptualization:** Bidyarthi Paul, Shuvashis Sarker.

**Data curation:** Bidyarthi Paul, Faika Fairuj Preotee, Shamim Rahim Refat.

**Formal analysis:** Shuvashis Sarker, Shamim Rahim Refat.

**Investigation:** Shuvashis Sarker, Shifat Islam.

**Methodology:** Bidyarthi Paul, Shuvashis Sarker, Shamim Rahim Refat.

**Supervision:** Tashreef Muhammad, Mohammad Ashraful Hoque.

**Validation:** Shifat Islam, Shahriar Manzoor.

**Writing – original draft:** Bidyarthi Paul, Shuvashis Sarker.

**Writing – review & editing:** Shifat Islam, Tashreef Muhammad.

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
