## [Decision Letter · Decision Letter 0]

20 Oct 2025

PONE-D-25-31659

ANCHOLIK-NER: A Benchmark Dataset for Bangla Regional Named Entity Recognition

PLOS ONE

Dear Dr. Muhammad,

Thank you for submitting your manuscript to PLOS ONE. After careful consideration, we feel that it has merit but does not fully meet PLOS ONE’s publication criteria as it currently stands. Therefore, we invite you to submit a revised version of the manuscript that addresses the points raised during the review process.

We look forward to receiving your revised manuscript.

Kind regards,

Matteo Bodini, Ph.D.

Academic Editor

PLOS ONE

Journal Requirements:

3. Thank you for uploading your study's underlying data set. Unfortunately, the repository you have noted in your Data Availability statement does not qualify as an acceptable data repository according to PLOS's standards.

Additional Editor Comments:

The manuscript presents a valuable and technically sound contribution to Bangla regional NER. For acceptance, the authors should include entity-wise F1 scores, per-dialect confusion matrices, and comparisons with existing Bangla NER datasets or LLM-based baselines. An error analysis and confirmation of full data availability are also required. Minor improvements to the abstract, figures, and language are recommended.

Reviewer's Responses to Questions

**Comments to the Author**

1. Is the manuscript technically sound, and do the data support the conclusions?

Reviewer #1: Yes

Reviewer #2: Yes

Reviewer #3: Yes

2. Has the statistical analysis been performed appropriately and rigorously?

Reviewer #1: Yes

Reviewer #2: Yes

Reviewer #3: Yes

3. Have the authors made all data underlying the findings in their manuscript fully available?

Reviewer #1: No

Reviewer #2: Yes

Reviewer #3: Yes

4. Is the manuscript presented in an intelligible fashion and written in standard English?

Reviewer #1: Yes

Reviewer #2: Yes

Reviewer #3: Yes

5. Review Comments to the Author

Reviewer #1: The authors present a real and underexplored challenge NER in Bangla regional dialects—which is often ignored in NLP due to data scarcity and linguistic diversity.

Creation of ANCHOLIK-NER is a useful resource contribution for the Bangla NLP community.: The annotation process, pre-processing, and evaluation are well explained with detailed steps and illustrative figures. They use standard BIO tagging and Cohen’s Kappa for inter-annotator agreement, enhancing transparency.

The evaluation with three transformer-based models (Bangla BERT, Bangla BERT Base, and Multilingual BERT) provides a sound baseline for future benchmarking.

While the dataset is new, methodologically, the work is largely a synthesis of established practices: standard BIO annotation, off-the-shelf BERT models, and conventional metrics.

The paper overstates novelty in model usage, even though the models are not fine-tuned innovatively or modified for dialectal handling (e.g., no dialect-aware pretraining or adapters). Authors should also try some LLMs, so they can show performance with discriminative models like BERT and generative models like llama, mistral as in https://link.springer.com/article/10.1007/s10115-024-02321-1, which is a comparative stuy.

No error analysis or detailed per-entity-type performance is included. For instance, which entity types (e.g., FOOD, ROLE, REL) are harder for the models across dialects? This is essential given the known imbalance.

Performance variance across regions is acknowledged (e.g., Chittagong underperforms), but no linguistic or statistical explanation is provided. Lack of baseline comparison against prior Bangla NER datasets like B-NER, BNLP, or IndicNER makes it difficult to quantify improvements.

The class distribution is highly skewed but no strategies (e.g., re-weighting, data augmentation, few-shot adaptation) are explored to address this.

There is no discussion on token overlap across dialects. Are some dialects lexically closer to Standard Bangla, making the task easier?

No attempt is made to adapt or augment models for dialect-specific tokens, even though dialect lexical gaps (e.g., merged/missing tokens) are acknowledged.

Use of multilingual BERT for dialect variation is questionable, as it is not tuned for intra-language dialectal variance, which limits its effectiveness.

The manuscript suffers from verbose repetition, and in places, casual phrasing like "going strong" or "real world variability" weakens the scientific tone.

Figures and Tables are numerous and informative but not always referenced or discussed sufficiently in the text.

Typographic errors (e.g., “Bangla Bert” instead of “Bangla BERT”) and inconsistent punctuation should be cleaned up.

Identify common failure modes per dialect or entity class. Is there confusion between FOOD vs. OBJ? LOC vs. ORG? Compare model performance with existing Bangla NER datasets if exists or use LLM based baselines to contextualize results.

Explore re-weighting or data augmentation to address class imbalance.

report entity-wise F1 scores and per-dialect confusion matrices.

overall, the dataset is a meaningful contribution to Bangla NLP, but the methodological novelty is limited, and the evaluation lacks analytical depth.

Reviewer #2: This is a great contribution to the researchers that are using language models in their research. Moreover, with LLMs and GenAI continuous developing, this NER dataset will help the models develop more accordingly, resulting in more accessibility for those in regions that use dialets. The authors could have emhasized these implications more in their conclusion. Thank you for the great work!

Reviewer #3: Dialectal variation is a core weakness of LLMs. Modern LLMs (even multilingual ones) still perform poorly on regional dialects. Fine-tuning or building benchmarks like ANCHOLIK-NER helps identify and quantify those weaknesses systematically.

This study makes a valuable contribution to the field for an important set of dialects. The authors identify the key problem, namely that existing Bangla NER models are trained on small or synthetic datasets, resulting in poor performance in many real-world contexts. The paper also serves as a model to follow for other sets of dialects.

The abstract should present a summary of other results, something along the lines of what is found on page 22 (“The results show that Bangla BERT performed best in the Mymensingh region, achieving the highest F1-score of 82.268% at epoch 20. In Barishal, it also performed well, reaching an F1-score of 81.481% at epoch 20. Sylhet and Noakhali showed moderate performance, with Sylhet achieving a peak F1-score of 78.754% at epoch 20, and Noakhali reaching 78.497% at epoch 20. The Chittagong region, however, showed relatively lower performance compared to the other regions, with the highest F1-score of 75.307% at epoch 20. Overall, Bangla BERT demonstrated strong performance, with its highest F1-scores observed in Mymensingh and Barishal.”)

Minor language errors exist (“it’s” instead of “its”, for example) but overall the paper is well written, carefully organized and systematically presented. Statistical analyses are appropriate. Conclusions are supported by the data.

Figures 6 and 8 are too small to read properly.

6. PLOS authors have the option to publish the peer review history of their article (what does this mean?). If published, this will include your full peer review and any attached files.

Reviewer #1: No

Reviewer #2: No

Reviewer #3: No

---

## [Author Response · Author response to Decision Letter 1]

10 Dec 2025

Additional Editor Comments: The manuscript presents a valuable and technically sound contribution to Bangla regional NER. For acceptance, the authors should include entity-wise F1 scores, per-dialect confusion matrices, and comparisons with existing Bangla NER datasets or LLM-based baselines. An error analysis and confirmation of full data availability are also required. Minor improvements to the abstract, figures, and language are recommended.

Response: We appreciate the editor’s feedback. In the revised manuscript we have addressed all the issues.

Reviewer #1

Comment-1: While the dataset is new, methodologically, the work is largely a synthesis of established practices: standard BIO annotation, off-the-shelf BERT models, and conventional metrics. The paper overstates novelty in model usage, even though the models are not fine-tuned innovatively or modified for dialectal handling (e.g., no dialect-aware pretraining or adapters).

Response: We appreciate the reviewer’s observation. Our study does not aim to introduce new model architectures but rather to benchmark existing pretrained transformer models on Bangla regional dialects. This approach ensures reproducibility and isolates the dataset’s contribution to performance. Accordingly, we have revised the Introduction and Methodology section to clarify that pretrained models were used for fine-tuning and evaluation, emphasizing benchmarking over model innovation.

Comment-2: Authors should also try some LLMs, so they can show performance with discriminative models like BERT and generative models like llama, mistral as in https://link.springer.com/article/10.1007/s10115-024-02321-1, which is a comparative study.

Response: We thank the reviewer for this valuable suggestion. We agree that evaluating large generative LLMs (e.g., LLaMA, Mistral) would provide an interesting complementary perspective. However, conducting such experiments requires substantial computational resources and inference tokens, which were beyond the feasible scope of the present study. To maintain reproducibility, uniform training conditions, and consistent baseline comparison, we focused on fine-tuning openly available discriminative transformer models commonly used in Bangla NLP research.

We have now clarified this in the Limitations and Future Work section and added a note that benchmarking large generative LLMs on ANCHOLIK-NER is a promising direction for future expansion of the dataset’s benchmark suite.

Comment-3: No error analysis or detailed per-entity-type performance is included. For instance, which entity types (e.g., FOOD, ROLE, REL) are harder for the models across dialects? This is essential given the known imbalance.

Response: We thank the reviewer for this observation. We have now added a dedicated error analysis section that includes detailed per-entity F1-scores across all five dialects. This analysis clearly identifies which entity types are harder (e.g., ANI, ROLE, ORG) and which are easier (e.g., FOOD, LOC, REL), directly addressing imbalance-related performance differences.

Comment-4: Performance variance across regions is acknowledged (e.g., Chittagong underperforms), but no linguistic or statistical explanation is provided. Lack of baseline comparison against prior Bangla NER datasets like B-NER, BNLP, or IndicNER makes it difficult to quantify improvements.

Response: We appreciate the reviewer’s suggestion regarding baseline comparison. However, the cited datasets (B-NER, BNLP, and IndicNER) are designed for Standard Bangla and do not account for regional or dialectal variation. As our dataset focuses exclusively on Bangla regional dialects, direct comparison with Standard Bangla benchmarks would not be methodologically appropriate due to differences in domain, linguistic distribution, and annotation schema.

Comment-5: The class distribution is highly skewed but no strategies (e.g., re-weighting, data augmentation, few-shot adaptation) are explored to address this.

Response: We appreciate this suggestion. In response, we conducted an additional experiment using class-weighted loss on Bangla BERT to directly address the skewed class distribution. We have added a new subsection describing this approach along with updated entity-wise results and confusion matrices. This provides insight into how re-weighting affects minority entity types.

Comment-6: There is no discussion on token overlap across dialects. Are some dialects lexically closer to Standard Bangla, making the task easier?

Response: We thank the reviewer for this feedback. We have now added a discussion that examines lexical similarity across dialects and its impact on model performance. Dialects closer to Standard Bangla (e.g., Barishal, Mymensingh) show higher scores, while those with greater lexical divergence (e.g., Chittagong, Noakhali) exhibit more errors. This explanation has been incorporated into the revised manuscript.

Comment-7: No attempt is made to adapt or augment models for dialect-specific tokens, even though dialect lexical gaps (e.g., merged/missing tokens) are acknowledged.

Response: We thank the reviewer for pointing this out. The current study intentionally did not modify pretrained transformer models to handle dialect-specific lexical variations such as merged, missing, or regionally altered tokens. Our goal was to benchmark existing models on the ANCHOLIK-NER dataset in a controlled and reproducible manner, isolating the contribution of the dataset itself to model performance. We have clarified this choice in the Methodology section by noting that all models were fine-tuned using their default tokenizers and vocabularies. Additionally, we have added a discussion in the Conclusion and Future Work section acknowledging this limitation and highlighting future directions, including dialect-aware tokenization, vocabulary extension, and adapter- or LoRA-based fine-tuning to better capture region-specific lexical patterns.

Comment-8: Use of multilingual BERT for dialect variation is questionable, as it is not tuned for intra-language dialectal variance, which limits its effectiveness.

Response: We thank the reviewer for this insightful observation. To address this point and clarify our methodological rationale, we have added a new Discussion section. This section explains the inclusion of multilingual BERT as a cross-lingual baseline, interprets the performance differences across dialects, and discusses broader implications for dialect-aware LLMs and future research.

Comment-9: The manuscript suffers from verbose repetition, and in places, casual phrasing like "going strong" or "real world variability" weakens the scientific tone.

Response: We thank the reviewer for this valuable observation. We have thoroughly reviewed the manuscript to remove redundant expressions and improve academic tone throughout the text. Informal phrases such as “going strong” and “real world variability” have been replaced with formal equivalents.

Comment-10: Figures and Tables are numerous and informative but not always referenced or discussed sufficiently in the text. Typographic errors (e.g., “Bangla Bert” instead of “Bangla BERT”) and inconsistent punctuation should be cleaned up.

Response: We thank the reviewer for this valuable feedback. We have revised the manuscript to ensure that all Figures and Tables are now clearly referenced and discussed in the corresponding sections. In addition, all typographic inconsistencies, including capitalization errors such as “Bangla Bert,” and punctuation issues have been carefully corrected throughout the manuscript to improve readability and presentation quality.

Comment-11: Identify common failure modes per dialect or entity class. Is there confusion between FOOD vs. OBJ? LOC vs. ORG?

Response: We thank the reviewer for this encouraging feedback We have now analyzed all dialect-specific confusion matrices and identified recurrent failure modes. The revised manuscript includes a clear description of common misclassification patterns.

Reviewer #2

Comment-1: This is a great contribution to the researchers that are using language models in their research. Moreover, with LLMs and GenAI continuous development, this NER dataset will help the models develop more accordingly, resulting in more accessibility for those in regions that use dialects. The authors could have emphasized these implications more in their conclusion. Thank you for the great work!

Response: We thank the reviewer for their encouraging feedback and insightful suggestion. In the revised manuscript, we have expanded the conclusion to emphasize the broader implications of ANCHOLIK-NER for large language models (LLMs), generative AI, and linguistic inclusivity.

Reviewer #3

Comment-1: The abstract should present a summary of other results, something along the lines of what is found on page 22 (“The results show that Bangla BERT performed best in the Mymensingh region, achieving the highest F1-score of 82.268% at epoch 20. In Barishal, it also performed well, reaching an F1-score of 81.481% at epoch 20. Sylhet and Noakhali showed moderate performance, with Sylhet achieving a peak F1-score of 78.754% at epoch 20, and Noakhali reaching 78.497% at epoch 20. The Chittagong region, however, showed relatively lower performance compared to the other regions, with the highest F1-score of 75.307% at epoch 20. Overall, Bangla BERT demonstrated strong performance, with its highest F1-scores observed in Mymensingh and Barishal.”)

Response: We thank the reviewer for this helpful suggestion. In the revised manuscript, we have updated the Abstract to include a concise summary of region-wise F1-scores and comparative model performance, ensuring that key quantitative results are clearly presented upfront.

Comment-2: Minor language errors exist (“it’s” instead of “its”, for example) but overall the paper is well written, carefully organized and systematically presented. Statistical analyses are appropriate. Conclusions are supported by the data.

Response: We thank the reviewer for the positive feedback and kind remarks regarding the overall organization and presentation of our manuscript. All identified language errors, including instances such as the incorrect use of “it’s” instead of “its,” have been carefully corrected throughout the revised version.

Comment-3: Figures 6 and 8 are too small to read properly.

Response: We appreciate the reviewer’s observation. In the revised manuscript, Figures 6 and 8 have been resized and reformatted to ensure improved readability and visual clarity.

---

## [Decision Letter · Decision Letter 1]

28 Jan 2026

ANCHOLIK-NER: A Benchmark Dataset for Bangla Regional Named Entity Recognition

PONE-D-25-31659R1

Dear Dr. Muhammad,

We’re pleased to inform you that your manuscript has been judged scientifically suitable for publication and will be formally accepted for publication once it meets all outstanding technical requirements.

Kind regards,

Joanna Tindall, PhD

Staff Editor

PLOS One

Additional Editor Comments (optional):

Reviewers' comments:

Reviewer's Responses to Questions

**Comments to the Author**

1. If the authors have adequately addressed your comments raised in a previous round of review and you feel that this manuscript is now acceptable for publication, you may indicate that here to bypass the “Comments to the Author” section, enter your conflict of interest statement in the “Confidential to Editor” section, and submit your "Accept" recommendation.

Reviewer #2: All comments have been addressed

Reviewer #3: All comments have been addressed

2. Is the manuscript technically sound, and do the data support the conclusions?

Reviewer #2: Yes

Reviewer #3: Yes

3. Has the statistical analysis been performed appropriately and rigorously?

Reviewer #2: Yes

Reviewer #3: Yes

4. Have the authors made all data underlying the findings in their manuscript fully available?

Reviewer #2: Yes

Reviewer #3: Yes

5. Is the manuscript presented in an intelligible fashion and written in standard English?

Reviewer #2: Yes

Reviewer #3: Yes

6. Review Comments to the Author

Reviewer #2: The research team made efforts to address the feedbacks from the initial review. Through this process the paper is in a much better place and I fully recommend this paper to be accepted.

Reviewer #3: All my comments have been adequately addressed. The paper is ready for publication, in my view. The authors have been very responsive.

7. PLOS authors have the option to publish the peer review history of their article (what does this mean?). If published, this will include your full peer review and any attached files.

Reviewer #2: No

Reviewer #3: No

---

## [Editor Report · Acceptance letter]

PONE-D-25-31659R1

PLOS One

Dear Dr. Muhammad,

I'm pleased to inform you that your manuscript has been deemed suitable for publication in PLOS One. Congratulations! Your manuscript is now being handed over to our production team.

Kind regards,

on behalf of

Dr Joanna Tindall

Staff Editor

PLOS One